# Structural basis for species-selective targeting of Hsp90 in a pathogenic fungus

Luke Whitesell[1,2], Nicole Robbins[1], David S. Huang[3], Catherine A. McLellan[2], Tanvi Shekhar-Guturja[1], Emmanuelle V. LeBlanc[1], Catherine S. Nation[4], Raymond Hui[5], Ashley Hutchinson[5], Cathy Collins[1], Sharanya Chatterjee[6], Richard Trilles[3], Jinglin L. Xie[1], Damian J. Krysan[7], Susan Lindquist[2,8], John A. Porco Jr.[3], Utpal Tatu[6], Lauren E. Brown [3], Juan Pizarro [4] & Leah E. Cowen[1]

New strategies are needed to counter the escalating threat posed by drug-resistant fungi. The molecular chaperone Hsp90 affords a promising target because it supports survival, virulence and drug-resistance across diverse pathogens. Inhibitors of human Hsp90 under development as anticancer therapeutics, however, exert host toxicities that preclude their use as antifungals. Seeking a route to species-selectivity, we investigate the nucleotide-binding domain (NBD) of Hsp90 from the most common human fungal pathogen, *Candida albicans*. Here we report structures for this NBD alone, in complex with ADP or in complex with known Hsp90 inhibitors. Encouraged by the conformational flexibility revealed by these structures, we synthesize an inhibitor with >25-fold binding-selectivity for fungal Hsp90 NBD. Comparing co-crystals occupied by this probe vs. anticancer Hsp90 inhibitors revealed major, previously unreported conformational rearrangements. These insights and our probe's species-selectivity in culture support the feasibility of targeting Hsp90 as a promising antifungal strategy.

[1] Department of Molecular Genetics, University of Toronto, Toronto, ON M5G 1M1, Canada. [2] Whitehead Institute for Biomedical Research, Cambridge, MA 02142, USA. [3] Department of Chemistry, Center for Molecular Discovery, Boston University, Boston, MA 02215, USA. [4] Department of Tropical Medicine, School of Public Health and Tropical Medicine and Vector-Borne Infectious Disease Research Center, Tulane University, New Orleans, LA 70112, USA. [5] Structural Genomics Consortium, University of Toronto, Toronto, ON M5G 1L7, Canada. [6] Department of Biochemistry, Indian Institute of Science, Bangalore 560012, India. [7] Departments of Pediatrics and Microbiology/Immunology, Carver College of Medicine, University of Iowa, Iowa City, IA 52242, USA. [8] Department of Biology, Howard Hughes Medical Institute, Massachusetts Institute of Technology, Cambridge, MA 02139, USA. These authors contributed equally: Lauren E. Brown, Juan Pizarro, Leah E. Cowen. Deceased: Susan Lindquist. Correspondence and requests for materials should be addressed to L.E.C. (email: leah.cowen@utoronto.ca)

Fungal pathogens are major contributors to human morbidity and mortality, infecting billions of people worldwide and killing in excess of 1.5 million per year[1]. The most vulnerable are individuals with compromised immune systems, including those undergoing chemotherapy, recipients of solid organ or hematopoietic stem cell transplants, and those infected with HIV. *Candida albicans* is a common commensal member of the human mucosal microbiome, but it is also the most frequent cause of invasive fungal infections in hospitalized patients, with mortality rates for disseminated disease approaching 40%, even with current treatments[2]. The development of drugs with selective activity against fungi has been hampered by the close evolutionary relationship between fungi and their human hosts[3]. An armamentarium of only three broadly deployed drug classes currently exists, and the frequent emergence of resistance makes cure of invasive fungal infections often unachievable. New agents with non-cross-reactive modes of action and non-overlapping toxicities are urgently needed[4].

A promising strategy to enhance the efficacy of established antifungals and improve clinical outcome is the co-targeting of regulators of fungal stress responses[5]. We previously established that the evolutionarily ancient, highly conserved molecular chaperone Hsp90 promotes antifungal drug tolerance and the evolution of drug resistance in species of *Candida* and *Aspergillus*[6–8]. Hsp90 is essential in all eukaryotes where it regulates the form and function of diverse client proteins, many of which are key signal transducers[9]. Hsp90 enables drug resistance via multiple molecular mechanisms, most notably through the stabilization of the stress-activated protein phosphatase calcineurin and multiple components of Pkc1 mitogen-activated protein kinase (MAPK) cell wall integrity pathway[6,8,10,11]. Hsp90 also serves as the previously enigmatic thermal sensor that controls temperature-dependent morphogenetic switching in *C. albicans*[12], a key factor contributing to the organism's virulence[13,14]. In animal models, limiting Hsp90 function restores sensitivity in drug-resistant fungal pathogens, impedes the evolution of drug resistance, and leads to clearance of otherwise lethal infections[7,8,12,15]. Unfortunately, compromise of host Hsp90 function with human homolog-optimized drugs (currently being tested in patients with cancer) comes with prohibitive side effects in the context of systemic fungal infections[7,16]. Thus, while fungal Hsp90 serves as a promising therapeutic Achilles heel, the utility of targeting it depends on whether sufficient inhibitor selectivity can be achieved to avoid compromise of its human homolog[5,17].

The ability to inhibit nucleotide binding within the Bergerat fold present in Hsp90's N-terminal domain was initially established by two structurally unrelated macrocyclic natural products, the resorcinylic acid lactone radicicol and the benzoquinone ansamycin geldanamycin[18,19]. With the goal of disrupting Hsp90's unique role in enabling oncogenesis, multiple structure-guided anticancer drug development programs have yielded a diverse array of inhibitor chemotypes[20–22]. Utilizing a variety of high affinity binding modes, these compounds all mimic the unusual conformation adopted by ATP upon binding within the chaperone's nucleotide-binding domain (NBD) and inhibit ATP binding and hydrolysis, thereby blocking chaperone function[23].

Consistent with its evolutionarily ancient and essential role in maintaining eukaryotic protein homeostasis, Hsp90 is highly conserved across phylogenetic kingdoms. For bacterial, yeast and human Hsp90, an ordered three-state cycle of conformational changes associated with the hydrolysis of ATP is fundamental to the protein's chaperoning activity[24]. However, the equilibria between these states and the intrinsic ATPase activity of the isoforms are highly species-specific[25]. The disparity has been hypothesized to reflect "evolutionary tuning to meet the client protein and metabolic environment of an organism"[26]. An additional layer of conformational regulation is provided by a suite of co-chaperones and accessory proteins, which also varies in composition across species. These proteins physically associate with Hsp90 to modulate its client-protein binding and the conformational changes associated with its chaperoning cycle[27]. Thus, despite a high degree of primary sequence conservation, functionally important conformational differences have been demonstrated between the Hsp90 isoforms expressed in humans and the model fungus *Saccharomyces cerevisiae*[27]. To date, however, no Hsp90 structural studies have been performed in pathogenic fungi such as *C. albicans*.

Seeking to target potential species-dependent differences in conformation or flexibility within the ATP-binding pocket of Hsp90, we pursue an iterative chemo-structural approach to achieve fungal-selective inhibition. We first solve the structure of the *C. albicans* Hsp90 N-terminal NBD, alone and in complex with several known Hsp90 inhibitors. The insights gained helped guide synthesis of a fungal-selective probe molecule, CMLD013075 (**1**). This probe is then co-crystalized with the NBD to reveal a surprising, previously unreported binding-site rearrangement permitting accommodation of the ligand. The unique binding mode observed for our probe and the structural flexibility of the fungal NBD suggest a promising path to selectivity despite high conservation at the primary sequence level. Our findings provide proof-of-principle for the feasibility of selectively targeting fungal Hsp90, while the biological activities of our probe support the therapeutic potential of targeting this key molecular hub to counter the escalating problem of drug-resistant fungal infections[4].

## Results

**Unique features of *C. albicans* Hsp90.** To identify potentially exploitable differences in the Hsp90 expressed by a fungal pathogen vs humans, we determined the crystal structure of the *C. albicans* Hsp90 N-terminal domain, which includes the nucleotide-binding pocket targeted by all Hsp90 inhibitors in clinical development[28,29]. We obtained structures for the domain under three conditions: *apo* (unliganded), in complex with ADP, and in complex with the prototypical inhibitor radicicol (Fig. 1a, Supplementary Table 1). Despite the presence of different ligands, overlay of the three crystal structures showed an almost identical conformation for this domain with a root mean square deviation (r.m.s.d.) of 1.32 Å or less over all atoms (Fig. 1a and Supplementary Table 2). Nonetheless, localized structural differences between the apo and liganded complexes were observed in the loop between α-helices 4 and 5 (residues Gly97 to Thr104), a region on the edge of the ATP binding site (Supplementary Figure 1). A portion of this region also differed between the fungal protein and its human homolog, amid otherwise very similar structures (Fig. 1b, c, upper panels). In previously reported human *apo* structures, residues 104–111 adopt either an open (PDB: 1YES) or closed (PDB: 1YER) loop conformation; in the *C. albicans apo* structure, the corresponding region (94–101) adopts an open loop conformation (Fig. 1c, upper panel). Overall comparison between both human *apo* structures (open and closed) and the fungal N-terminal domain showed no major differences, with main-chain atom r.m.s.d. of 1.02 and 1.15 Å, respectively. Global structural similarity between fungal and human Hsp90 was also observed when comparing the chaperone-ADP and -radicicol complexes (Fig. 1b, middle and lower panels). Additionally, the N-terminal domain of *C. albicans* Hsp90 shares the same architecture at the nucleotide-binding site, which interacts with both ADP and radicicol through hydrogen-bond interactions at the same amino acids (Asp92 and Thr174, Fig. 1c, middle and lower panels).

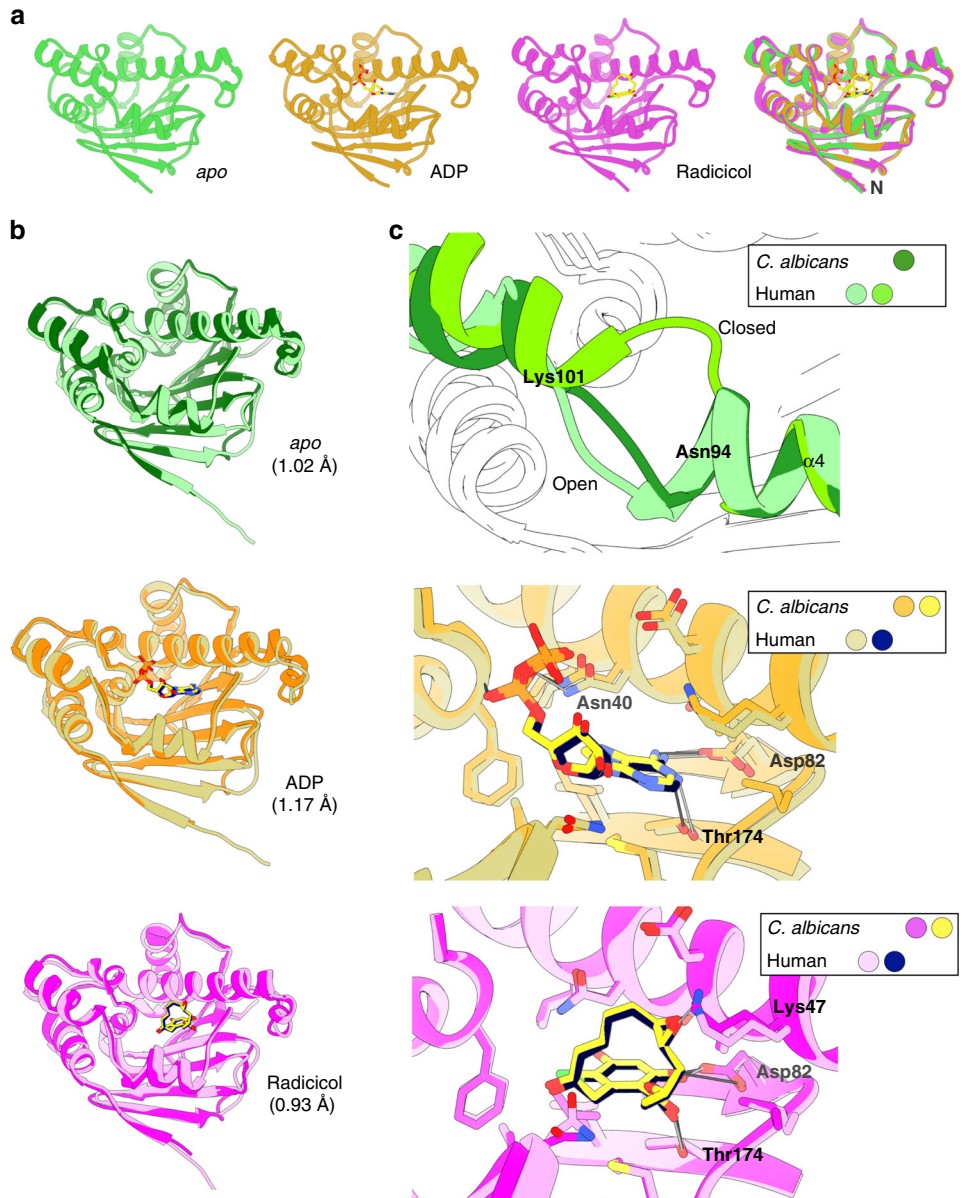

**Fig. 1** Structure of *C. albicans* Hsp90 nucleotide-binding domain (NBD) in *apo* and ligand-bound states. **a** Ribbon representation of crystal structures determined for *C. albicans* Hsp90 NBD in the non-ligand-bound or *apo* state (green), ADP-bound state (yellow) and in complex with radicicol (magenta). The ligands ADP and radicicol are represented as sticks and color-coded according to their heteroatom composition. An overlay of all three structures is also presented at the far right. **b** Ribbon representation of crystal structures determined for *C. albicans* NBD (darker shades) in the non-ligand-bound or *apo* state (top panel, green), ADP-bound state (middle panel, orange), and in complex with radicicol (lower panel, magenta). Structures for human Hsp90 NBD (lighter shades) are overlaid. The ligands ADP and radicicol are represented as sticks and color-coded according to their heteroatom composition. The main-chain atom r.m.s.d. for the superpositions is indicated in parentheses for each overlaid pair. **c** Detailed view of superpositions encompassing the region of the ligand-binding site. The top panel provides a detailed view of the region Asn94–Lys 101 in *C. albicans* and human *apo* structures, two human conformations are shown open (lighter green; PDB id. 1YES) and closed (darker green; PDB id. 1YER). The middle and bottom panels depict a detailed view of the residues which hydrogen-bond with ADP and radicicol, respectively, in both *C. albicans* and human Hsp90 NBD

Given the structural similarities between human and *C. albicans* Hsp90 NBD, we asked whether the core ATPase activity of purified full-length homodimers from each species would also be conserved. The dissociation constant ($K_d$) for ATP of *C. albicans* Hsp90 was similar to that previously reported for *S. cerevisiae* Hsp90 (125 vs 132 μM), but somewhat lower than that reported for human Hsp90 (240 μM; Supplementary Figure 2a, 2d). The ATPase activity of *C. albicans* Hsp90 obeyed Michaelis–Menten kinetics (Supplementary Figure 2b and 2c), yielding values for catalytic efficiency similar to that reported for *S. cerevisiae* ($11.6 \times 10^{-5}$ vs $15.6 \times 10^{-5}$) but more than twofold

higher than that reported for human Hsp90 ($4.6 \times 10^{-5}$; Supplementary Figure 2d). At the level of biochemical activity, therefore, a difference is apparent between *C. albicans* Hsp90 and its mammalian counterpart despite extensive sequence conservation within the NBD.

**Increased conformational flexibility in *C. albicans* Hsp90.** Seeking to reconcile apparent structural conservation with disparity in ATPase activity, we probed the conformational flexibility of the *C. albicans* NBD by solving structures of the domain

in complex with two clinically relevant Hsp90 inhibitors, SNX-2112 (Fig. 2a, Supplementary Table 1) and AUY922 (Fig. 2b, Supplementary Table 1). These compounds were chosen because previously reported human Hsp90 structures had shown distinct conformational changes in the region defined by Hsp90 residues 104–111 in co-crystals with these ligands[30]. While the human Hsp90-AUY922 complex shows an *apo*-like, closed-loop conformation, the human NBD in complex with SNX-2112 shows an extended α-helical conformation. This conformation exposes a hydrophobic pocket that has been previously characterized in human Hsp90[31]. Interestingly, compounds that induce a helical conformation of the 104–111 region show orthologue selectivity within the human Hsp90 family, binding poorly to Grp94 and Trap-1.

To correlate structural insights with the relative binding affinity for compounds, we used a fluorescence polarization (FP)-based competition approach that allows quantitation of ligand binding in assays utilizing purified NBD proteins or whole-cell lysates. Use of purified proteins allows determination of actual ligand dissociation constants for specific NBD[32]. Use of lysates interrogates overall ligand binding by Hsp90 in its native complexes with co-chaperones and in the case of human cells, the mixture of Hsp90 isoforms present in the lysate. Lysate experiments can only provide assay-dependent relative EC$_{50}$ values for inhibitors, but may be more relevant to selectivity at the whole-cell level. To avoid problems with auto-fluorescence typical of lysate, we synthesized a previously reported, red-shifted Cy3B-geldanamycin conjugate to use as the FP probe in both configurations of the assay[33]. Importantly, using this probe with purified human NBD protein yielded an inhibitory $K_i$ for geldanamycin consistent with previously reported values for full-length recombinant Hsp90α[34]. In our FP assay, SNX-2112 showed threefold higher affinity for the fungal than human Hsp90-binding domain (Fig. 2a, right panel and Supplementary Figure 3a, 3b). In lysates, the compound displayed even greater fungal selectivity, but the improvement may have resulted from the relatively poor affinity of SNX-2112 for Grp94 which is reported to be present in almost equimolar concentration to Hsp90 α/β in HEPG2 cells from which the human lysate was prepared (Supplementary Figure 3b and 3c)[35].

The structure of the *C. albicans* NBD–SNX-2112 co-crystal showed that the aminobenzamide and cyclohexanol portions of SNX-2112 occupy the adenosine region of the ATP-binding pocket, making similar polar contacts with the fungal Hsp90 as those seen in the human structure; overall both complexes showed considerable structural similarity (r.m.s.d. 1.0 Å, main-chain atoms) including almost identical poses for SNX-2112 (Fig. 2a). However, in the fungal NBD, the dihydroindazolone moiety of SNX-2112 induced a localized structural rearrangement of the Val93–Ser102 (103–112 in human Hsp90) region distinct from the one observed in the human complex structure. Unlike the extended α-helix observed for the human NBD in complex with SNX-2112, the *C. albicans* NBD adopted a bulged-out loop conformation with a shortening of α-helix 4 (Fig. 2a, right panel). A similar conformation was previously described for a complex of *T. brucei* Hsp90 NBD with SNX-2112 complex (PDB: 3OPD). Interestingly, the Hsp90 in this organism showed increased affinity for SNX-2112[36]. An overlay between *C. albicans* NBD *apo* and SNX-2112 structures further demonstrates the significance of the structural rearrangement within the Val93–Ser102 region upon binding to SNX-2112, as this region shows the largest differences between both structures (Fig. 2c and Supplementary Figure 1). The plasticity of the *C. albicans* Hsp90 ATP-binding site appears to be an important element in understanding differences in binding affinity for *ortho*-benzamide-type Hsp90 inhibitors like SNX-2112, due to the highly localized nature of the

structural differences between the liganded *C. albicans* and human chaperones, and between the *C. albicans* liganded and *apo* structures.

In contrast to SNX-2112, the structure of *C. albicans* Hsp90 NBD in complex with AUY922 was different from all previously reported structures for an Hsp90 of any species, revealing a large structural rearrangement that extended beyond the nucleotide-biding site (Fig. 2b–d). This fungal co-crystal structure contrasts with a previously reported human NBD-AUY922 complex structure, which adopts a closed-loop *apo* conformation (PDB: 2VCI)[37]. Remarkably, the large structural rearrangements observed in the *C. albicans* AUY922 complex were accompanied by very high binding affinity for the compound, although less than that observed for the human NBD (Fig. 2b, right panel and Supplementary Figure 3b). As with SNX-2112, the high affinity of AUY922 for human Hsp90 NBD also appears to have been obscured somewhat in HEPG2 lysate by the lower affinity of Grp94 for this compound (Supplementary Figure 3b and 3c). An overlay of the fungal and human complexes highlights the extent of the structural changes associated with AUY922 binding to *C. albicans* NBD (Fig. 2b and Supplementary Figure 1, r.m.s.d. 3.8 Å, main-chain atoms). The observed conformational change extends beyond the nucleotide-binding site to include the N-terminal β-strand (β1) and helix (α1), and the lid subdomain comprising residues 94–126 (including helices α5 and α6) (Figure S4a). The loss of β1 also leads to a shortening of the adjacent strand (β6) and repositioning of the loop connecting β5-β6 (Supplementary Figure 4a).

Despite major structural differences between the *C. albicans* and human Hsp90 complexes with AUY922, the ligand maintains the majority of polar and non-polar contacts in each, adopting nearly identical orientations with the exception of the morpholine substituent (Fig. 2b, Supplementary Figure 4a and 4b). In complex with human Hsp90α, the morpholine points away from the chaperone into the solvent, engaging in limited interactions with the host protein at residues Thr109 and Gly135. In the *C. albicans* complex, the heterocycle is flipped ~180° towards the protein core, engaging in contacts with residues Asn95, Gly124 and Phe123 (Supplementary Figure 4b). The conformational changes induced by AUY922 binding in *C. albicans* Hsp90 suggest a far greater degree of conformational flexibility for the fungal chaperone compared to its human homolog.

To better understand the roles of the NBD's structural elements in ligand-binding, we overlaid the *C. albicans* Hsp90 *apo* and AUY922 structures (Fig. 2c, right). As previously noted, the *C. albicans* NBD-AUY922 complex showed a large conformational change leading to major overall differences with respect to the *apo* structure (r.m.s.d. 3.47 Å, all atoms). Upon AUY922 binding, residues 94–101 adopt a helical conformation that fuses helices 4 and 5 (Fig. 2d). This fused helical element peels the lid region away from the ATP-binding site to accommodate positioning of AUY922's morpholine ring. This repositioning is accompanied by disappearance of the N-terminal β-strand (β1), affecting β6 and its preceding loop. Despite this clear change, the rest of the protein's structure in both the *apo* and AUY922 complexes is almost identical. The portion of the *C. albicans* NBD which interacts with the base moiety of ATP, defined by residues from the β-sheet (β2 to β5) and helices α2, α3, α8, and α9, is extremely rigid. Furthermore, this rigid region showed no significant differences across all of the co-crystal structures we solved for the fungal NBD (Supplementary Figure 1). Considered together, our structural observations define the conformationally mobile portions of *C. albicans* NBD that participate in the binding of specific ligands. Most importantly, the magnitude of structural reorganization observed upon binding of AUY922 indicated a much higher degree of conformational flexibility in the fungal than human Hsp90 NBD.

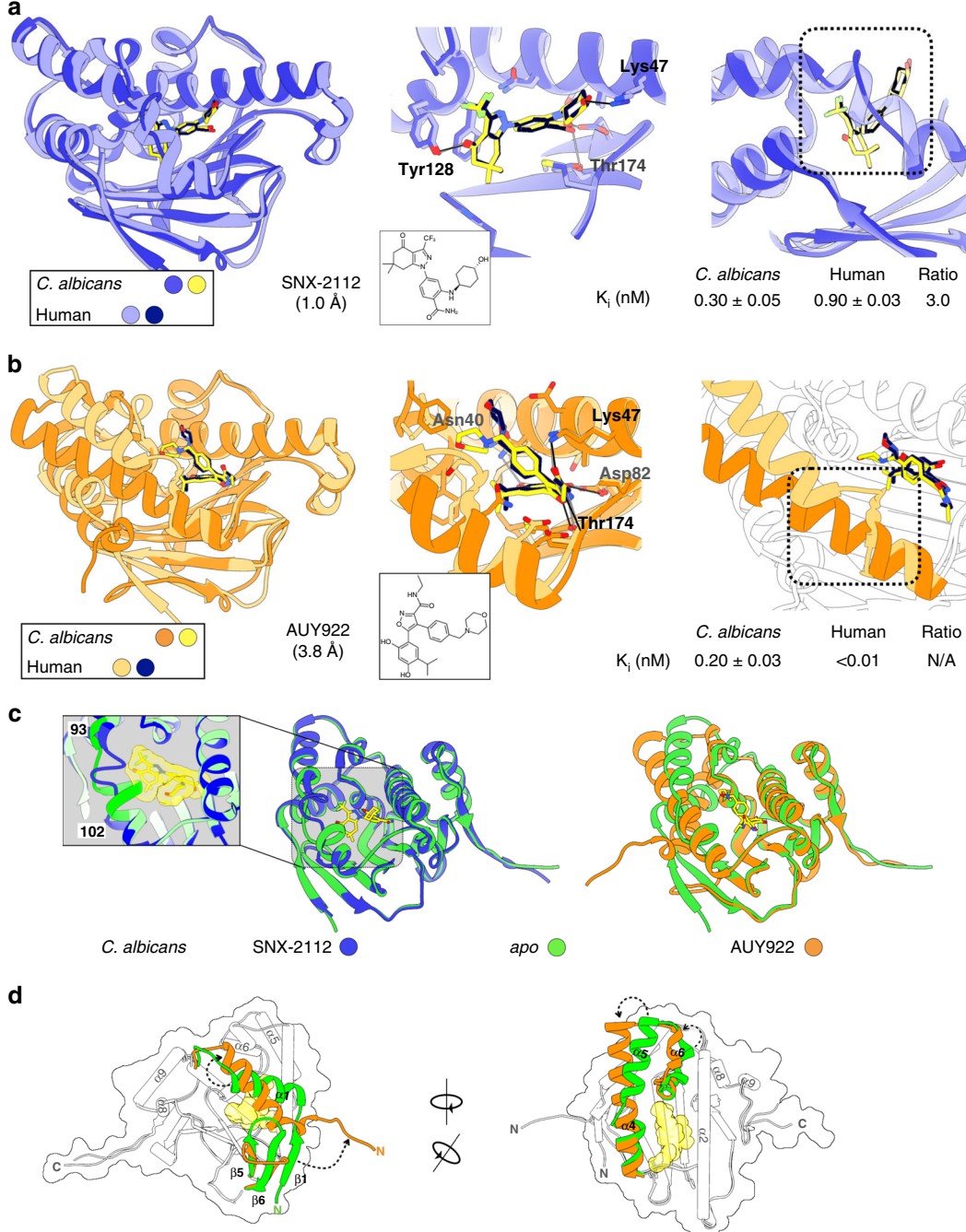

**Fig. 2** Comparison of inhibitor-bound structures for *C. albicans* vs human Hsp90 NBD. **a** Detailed view of the binding mode of SNX-2112 to *C. albicans* Hsp90 nucleotide-binding domain (NBD) (dark blue), overlaid with a human Hsp90α NBD structure in complex with SNX-2112 (light blue; PDB id. 4NH7). Several of the *C. albicans* Hsp90 residues interacting with the compound are identified, and the conformation of the 93–102 region is indicated by a dotted rectangle. Binding affinities for the compound were determined by equilibrium competition FP assay using purified human and fungal NBDs. $K_i$ values are indicated below the structures. The mean ± SEM of three independent experiments is presented. **b** Detailed view of the binding mode of AUY922 to *C. albicans* Hsp90 NBD (orange), overlaid with a human Hsp90α NBD structure in complex with the same compound (light orange; PDB id. 2VCI). Several *C. albicans* Hsp90 residues interacting with the compound are identified. The helical conformation of the 93–102 region is indicated by a dotted rectangle. Hsp90 NBD binding affinities were determined as described for panel a. The mean ± SEM of three independent experiments is presented. **c** Inhibitor-induced conformational changes in *C. albicans* NBD. Ribbon representation of *C. albicans* Hsp90 NBD bound to Hsp90 inhibitors SNX-2112 (left image) and AUY922 (right image). *Apo* structure (green) is overlaid. Compounds are represented as sticks and color-coded according to heteroatom composition. A detailed view of the conformational differences in the 93–102 region between *apo* and SNX-2112 bound crystal structures is provided; SNX-2112 is shown as sticks and a semitransparent yellow surface. **d** Views from different angles of the *C. albicans* Hsp90 NBD in its *apo* and AUY922-bound states. Overlays of the crystal structures are shown in cartoon/ribbon representation. Secondary structure elements are color-coded to highlight the conformational changes associated with AUY922 binding; dotted arrows indicate the direction of the movement of secondary structure elements. The outline of the *apo* surface is indicated by a black line, and the AUY922 position is indicated as a semitransparent yellow sphere representation. Source data for the determination of all inhibitory constants are provided as a source data file.

**Semi-synthesis of fungal-selective Hsp90 inhibitors**. Disparity between the ligand-induced conformational changes we identified in fungal vs human NBDs encouraged further work to synthesize fungal-selective inhibitors. To identify the most promising scaffold from which to begin, we tested a panel of Hsp90 inhibitor chemotypes for their ability to inhibit fungal growth. We used a strain of *S. cerevisiae* in which the genes encoding both isoforms of its endogenous Hsp90 (*HSP82* and *HSC82*) have been disrupted and growth is dependent upon *C. albicans* Hsp90 expressed from a plasmid. The natural product radicicol was the only Hsp90 inhibitor found to significantly inhibit growth of our engineered fungal strain (Supplementary Figure 5a, 5b). Our structural studies had demonstrated considerable ligand-induced flexibility within the NBD of fungal Hsp90, including an ability of the fungal, but not human, NBD to significantly rearrange to accommodate AUY922, an inhibitor which conserves the resorcinylic substructure of radicicol. As an approach to exploit this apparent flexibility, we pursued the introduction of steric bulk to radicicol via oxime derivatization (a strategy previously employed for metabolic stabilization of the compound)[38,39]. We began by investigating the binding to *C. albicans* vs human Hsp90 of KF58333, a previously described radicicol oxime (Fig. 3a).

To monitor relative Hsp90-binding affinities, we again used an FP-based competition approach in whole-cell lysates. We observed potent binding for KF58333, but nearly equivalent $EC_{50}$ values for both human and *Candida* Hsp90 (7–10 nM) (Fig. 3b and Supplementary Table 3). Presumably the oxime substituent in KF58333, which had been optimized to preserve activity in humans, is not sufficiently bulky to hinder binding of the compound. To address this, we next generated a small set of known (**2**)[40,41], (**8**) (USPTO Patent Application 20100292218) and novel oxime analogs (**3**–**7**) through the direct chemical modification of radicicol[42]. The compounds were prepared using standard conditions with commercially available alkoxyamines varying in size and substitution (Fig. 3c and Supplementary Methods). Also included in the set was an analog, CMLD013075 (**1**) derived from the structurally related *des*-chlorinated natural product monocillin I (Fig. 3c), which is roughly equipotent to radicicol as an Hsp90 inhibitor in plants[43]. By FP assay, we found a moderate (1.4- to 4.6-fold) gain in fungal selectivity for several of our new oxime analogs in binding Hsp90 in whole-cell extracts prepared from *C. albicans* (SC5314) compared to HEPG2, a human cancer cell line (Fig. 3b). Although our design efforts were inspired primarily by structural and biological data pertaining to radicicol, we serendipitously discovered that the most selective inhibitor was monocillin-derived CMLD013075. This compound exhibited superior fungal selectivity over all other analogs tested by FP lysate assay, with an $EC_{50}$ of ~500 nM in *C. albicans* lysate translating to a normalized 4.6-fold selectivity factor (Fig. 3b and Supplementary Table 3). Overall, our compound series supports a model in which the steric bulk of larger oximes can limit binding to the less flexible human Hsp90 NBD, yet still maintain, if not improve, fungal Hsp90-binding activity.

A critical consideration in the development of an effective antifungal is not just target binding, but also access to the target across microbial cell membranes and cell walls. Therefore, we evaluated the potency and selectivity of oxime analogs using drug susceptibility assays. We used strains of *S. cerevisiae* where both endogenous genes encoding isoforms of Hsp90 have been disrupted and growth is solely dependent upon *C. albicans* or human Hsp90 isoforms. All our oxime derivatives, including CMLD013075, displayed greater potency in limiting fungal proliferation than geldanamycin and KF58333 (Fig. 3d). Consistent with selectivity findings by FP, CMLD013075 was more active against a yeast strain engineered to express *C. albicans* Hsp90 than isogenic strains expressing either human

Hsp90α or Hsp90β proteins. The converse was true of radicicol (Fig. 3e).

**Hsp90 residues key to the fungal selectivity of CMLD013075**. To identify molecular contacts contributing to the relative selectivity of CMLD013075 for *C. albicans* Hsp90, we solved a co-crystal structure of the compound bound within the protein's N-terminal domain (Fig. 4a). Structural analysis revealed a substantial conformational rearrangement of the NBD associated with binding of the derivative, with an r.m.s.d. of 1.86 Å upon comparison to a radicicol co-crystal structure (Supplementary Table 2). Conformational rearrangement was associated with an inhibitory constant ($K_i$) by FP assay of 7.7 nM for binding of CMLD013075 to the fungal NBD and selectivity over binding to the NBDs of human Hsp90 and Grp94 of ~28- and ~5-fold respectively (Fig. 4c). The binding affinity of radicicol could not be assessed by FP due to its Michael reactivity and inactivation of the compound by thiol-reducing agent in the assay buffer[38,39]. The compound-induced conformation involved major movements of the N-terminal β-strand and repositioning of the lid subdomain that included α-helices 1, 5–7 and their linking loops (Supplementary Figure 1 and 6). These structural changes are accompanied by a partial remodeling of the ATP-binding site to accommodate the bulkier *p*-methoxybenzyl moiety of this compound (Fig. 4b and Supplementary Figure 8a, S8b). The macrocyclic portion of CMLD013075 occupies a position identical to that observed in the radicicol complex structure, with the oxime-linked *p*-methoxybenzyl group participating in several hydrophobic interactions as well as an apparent N-H⋯π bond to Asn40 (Fig. 4d).

Given the significant structural rearrangements associated with binding of CMLD013075 and AUY922 to *C. albicans* Hsp90, we next sought to directly compare these changes with respect to each other and to the *apo* structure (Supplementary Figure 6). Both the CMLD013075 and AUY922 complexes show similar changes in several secondary structure elements, such as the disappearance of the N-terminal β-strand (β1), the extension and tilting of helix α1, and the displacement of the lid subdomain (Supplementary Figure 6). However, in the CMLD013075 complex residues 94–101 adopt a bulged-in loop conformation, associated with a movement of the lid subdomain that appears essential to accommodate the *p*-methoxybenzyl group within the ATP-binding site (Supplementary Figure 8b). This conformational change differs considerably from the AUY922-induced structural rearrangement that involved the fusion of helices 4 and 5, and repositioning of the lid subdomain to accommodate AUY922's morpholine ring (Supplementary Figure 6).

To rationalize the enhanced selectivity of CMLD013075 for fungal Hsp90, we evaluated each amino acid residue that differed between the *C. albicans* and human NBD and assessed the individual contributions of these residues to local conformation by comparing their H-bond networks. In this way, we identified several residues within the fungal NBD that might affect the protein's conformation (Fig. 4e). We tested their importance by generating three *C. albicans* Hsp90 variants with the native fungal residues changed to their corresponding human residues (Fig. 5a, b). The first variant was a single substitution, replacing fungal threonine with human glutamine at position 12 (Thr12Gln) located between the N-terminal β-strand and α-helix. This substitution was predicted to affect the flexibility of the Gly94-Gly100 loop linking helices 4 and 5, thus potentially limiting movement of the lid region and N-terminus seen in the CMLD013075 complex structure. The second variant involved two amino acid changes within α-helix 7 (Leu130Ala and Phe131Tyr) which were predicted to impact contact with the

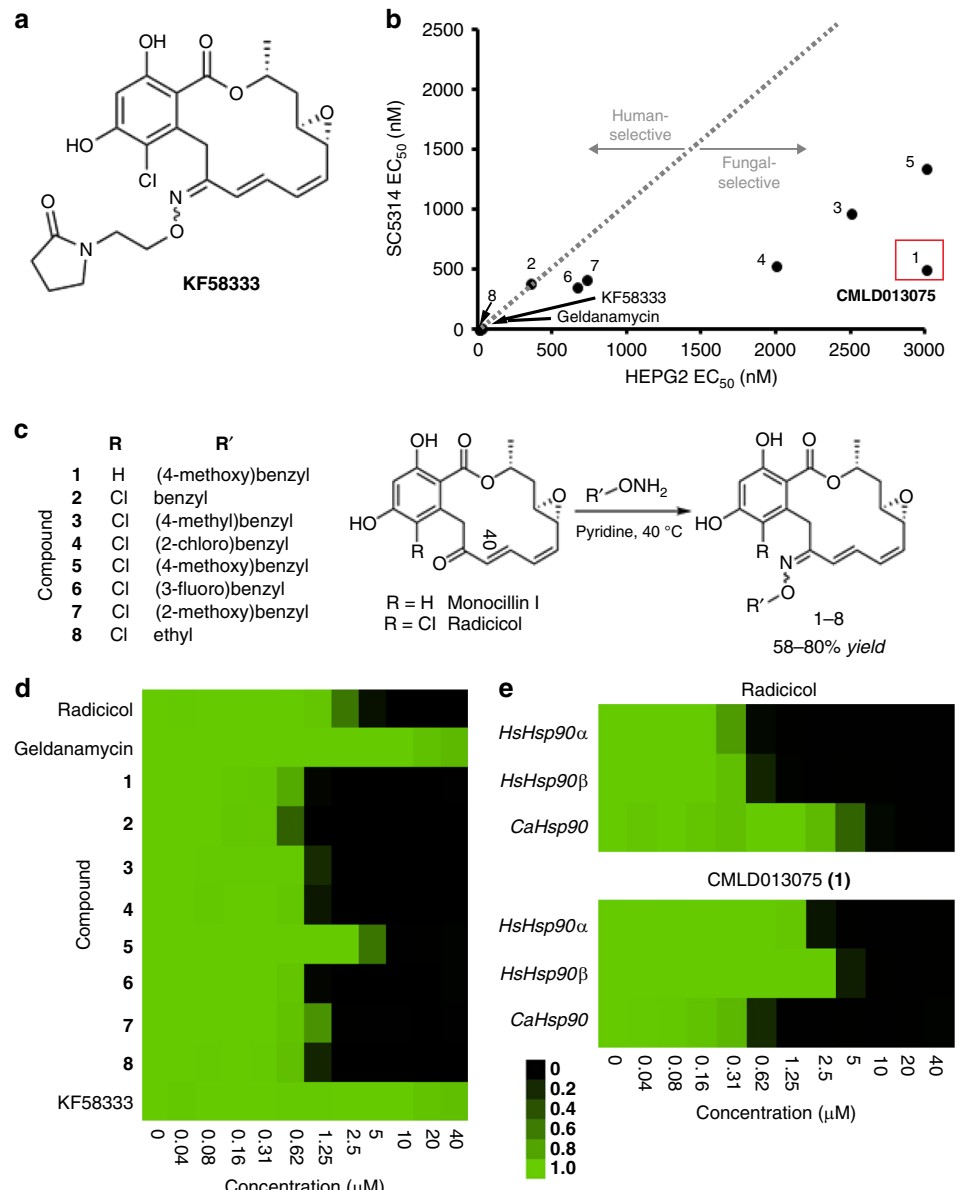

**Fig. 3** Synthesis and characterization of fungal-selective Hsp90 inhibitors. **a** Structure of previously reported anticancer radicicol oxime KF58333. **b** Plot of relative Hsp90-binding affinity for compounds determined in cytosolic extracts prepared from human (HEPG2) vs fungal (SC5314) cells. The relative binding affinities for compounds were determined in cytosolic extracts prepared from human vs fungal cells by fluorescence polarization (FP) assay and are indicated as normalized $EC_{50}$ values. A red box highlights the oxime derivative CMLD013075 (**1**) in our panel of analogs which demonstrated the greatest combination of potency and selectivity for fungal Hsp90. $EC_{50}$ values were determined by variable slope 4-parameter curve fitting of triplicate determinations at each concentration tested ($R^2 > 0.95$, all curves). All fungal values were normalized by the ratio of geldanamycin $EC_{50}$s measured in human vs fungal lysate. The experiment was repeated once with quantitatively similar results. Source data for the determination of $EC_{50}$ values are provided as a source data file. **c** Summary of R-group substitutions and scheme for the synthesis of monocillin-derived and radicicol-derived oxime analogs **1–8**. **d** Susceptibility testing of Hsp90 inhibitor panel using a strain of *S. cerevisiae* engineered to express *C. albicans* Hsp90 as its sole source of Hsp90. Relative growth inhibition by each compound over a twofold dilution series of concentrations is displayed in heat-map format. Each colored box represents the mean of duplicate determinations. Color scale for relative growth is provided in lower right of panel, green: no inhibition to black: complete inhibition. The experiment was repeated once to confirm results. **e** Inhibitor sensitivity of isogenic *S. cerevisiae* strains engineered to express the indicated *HSP90* genes as their sole source of Hsp90. Relative growth inhibition by radicicol (top) or oxime derivative CMLD013075 is displayed in the same manner and using the same color scale as in **d**. The experiment was repeated as an independent biological replicate to confirm results.

*para*-methoxybenzyl substituent. The final variant also involved two amino acid substitutions, Lys158Ser and Thr162Arg, located within β-strand 6. This strand is adjacent to β-strand 1 in the Hsp90 N-terminal β-sheet. To experimentally test the importance of these residues in providing fungal selectivity, we constructed strains of *S. cerevisiae* in which each of these variants constituted the sole source of Hsp90, and then compared the antifungal activity of radicicol and CMLD013075 against these strains. The *C. albicans* Hsp90 Thr12Gln or composite Lys158Ser/Thr162Arg substitutions did not alter the potency of either compound; radicicol remained less potent and CMLD013075 remained more potent against strains expressing these variants than against strains carrying the human isoforms (Fig. 5c). In contrast, the composite Leu130Ala/Phe131Tyr mutation increased potency of

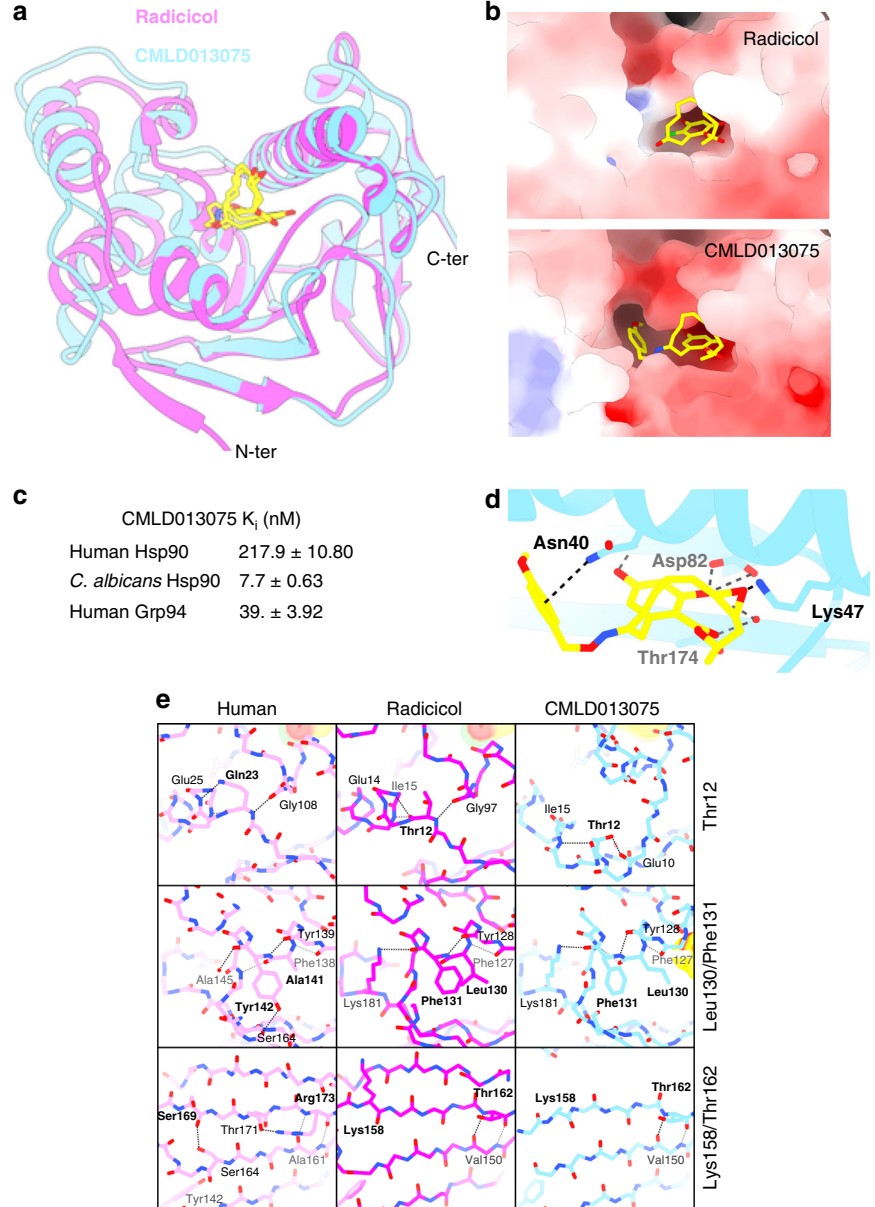

**Fig. 4** Structure of Hsp90 nucleotide-binding domain (NBD) in complex with CMLD013075. **a** Ribbon representation of a 2.6 Å resolution crystal structure of CMLD013075 (blue) bound to *C. albicans* Hsp90 NBD. Radicicol from a separate *C. albicans* co-crystal (magenta) is overlaid. The ligands are represented as sticks and color-coded according to their heteroatom composition. **b** Detailed view of the binding mode of radicicol (top) and CMLD013075 (bottom) to *C. albicans* Hsp90. A surface representation is provided and color-coded by electrostatic potential (−10 to 10kT/e blue to red color ramp). **c** Binding affinities for CMLD013075 were determined by equilibrium competition FP assay using purified human Hsp90, fungal Hsp90, and human Grp94 NBDs and are indicated as $K_i$ values. Assays were repeated in three independent experiments. The mean ± SEM is presented. Source data for the determination of binding affinities are provided as a source data file. **d** Polar contacts for the resorcinol-containing compound CMLD013075 in complex with *C. albicans* Hsp90 NBD. Protein residues engaged in H-bonds or cation–π interactions are labeled, and those interactions shown as dashed black lines. **e** H-bond networks of the residues selected for mutagenesis (bold residue names). Two fungal Hsp90 co-crystal structures are shown, radicicol (magenta), and CMLD013075 (cyan). For comparison, a human structure in complex with radicicol is also displayed (PDB id. 4EGK).

radicicol while decreasing potency of CMLD013075 relative to wild-type *C. albicans* Hsp90 (Fig. 5c). In fact, these substitutions in *C. albicans* Hsp90 resulted in even less sensitivity to CMLD013075 than observed upon expression of human Hsp90 isoforms, highlighting the importance of these amino acids in conferring fungal selectivity to the oxime.

**Species-selective activity of CMLD013075 in culture**. We focused on CMLD013075 for further biological characterization,

because it possessed the most potent and selective activity in whole-organism assays (Fig. 3d, e). We began by measuring inhibition of *C. albicans* growth in culture. Both radicicol and CMLD013075 showed comparable single-agent activity against a laboratory strain of *C. albicans* (SN95) as well as a more drug-resistant clinical isolate (CaCi-2) from an HIV-infected patient recovered early in the course of persistent *Candida* infection (Fig. 6a)[6].

Next, we investigated the fungal selectivity of our Hsp90 inhibitors using standard microplate-based dye-reduction assays

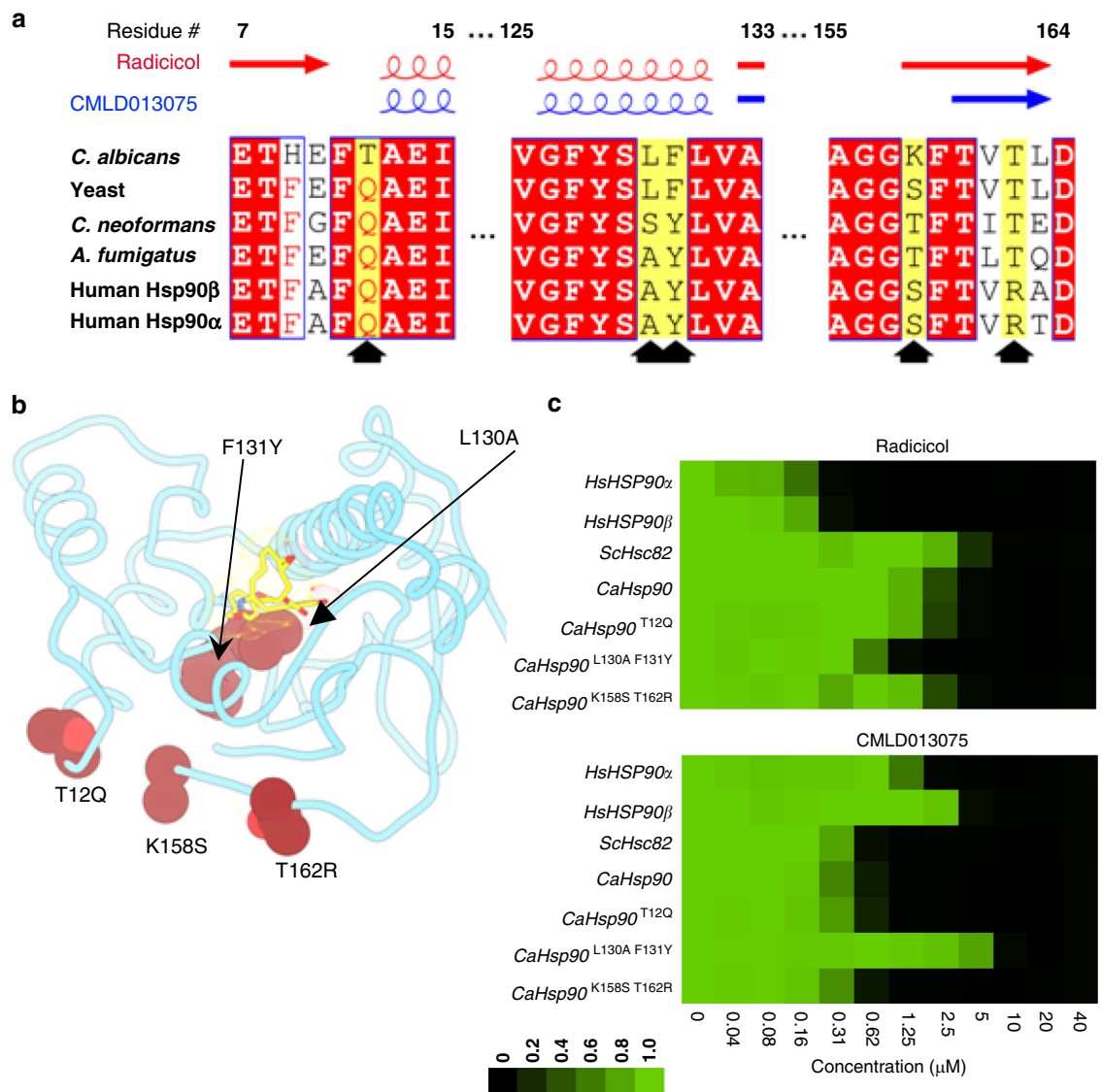

**Fig. 5** Identification of key residues for fungal selectivity. **a** The locations of secondary structural features observed in the respective radicicol and CMLD013075 co-crystal structures presented in Fig. 4a are mapped above the primary sequence alignments. Red shading indicates areas of complete sequence conservation. Black arrowheads indicate the position of the amino acids (shaded in yellow) that were changed for experiments involving mutagenesis. Positions with variation are indicated in red letters when a particular amino acid is most common and black letters when the position is variable. **b** Localization of residues mutated to examine effects on compound binding. Altered amino acids are represented as brick-colored spheres in the *C. albicans* NBD-CMLD013075 co-complex structure. Side chains of each residue are depicted. The ligand is represented as sticks and color-coded according to heteroatom composition. Light red spheres represent oxygen atoms. **c** Inhibitor sensitivity of isogenic *S. cerevisiae* strains engineered to express the indicated wild-type and mutant Hsp90 proteins as their sole source of Hsp90. Relative growth inhibition by twofold serial dilutions of radicicol (top) or oxime derivative CMLD013075 (bottom) is displayed in the same manner as in Fig. 3. Each colored box represents the mean of duplicate determinations. Color scale for relative growth is provided, green: no inhibition to black: complete inhibition. The experiment was repeated as an independent biological replicate to confirm results

to measure concentration-dependent mammalian cytotoxicity. CMLD013075 was >10-fold less toxic to human HEPG2 liver cancer cells than the non-selective parent compound radicicol, and >50-fold less toxic than ganetespib, a resorcinylic clinical inhibitor (Fig. 6b). Cytotoxicity was also tested against Raw264.7, mouse cancer cells of macrophage lineage, where again CMLD013075 was much less toxic than radicicol or ganetespib (Fig. 6b). We also examined the toxicity of CMLD013075 and several known Hsp90 inhibitors to non-cancer cells, i.e., the non-tumorigenic mouse fibroblast line 3T3 (Supplementary Figure 7a). Both radicicol and CMLD013075 were less cytotoxic to 3T3 cells than HEPG2 cells by at least 10-fold and both showed a plateau in

their dose–response curves consistent with the presence of a non-proliferating, relatively inhibitor-insensitive population in the contact-inhibited 3T3 line.

As an explanation for the greater activity of N-terminal Hsp90 inhibitors against cancers compared to normal tissues, it has been reported that Hsp90 in cancer cells resides predominantly in an activated hetero-complex form with higher inhibitor affinity[44,45]. To investigate whether the binding of CMLD013075 might differ in lysate from non-cancer cells and HEPG2 cells, we performed FP assays with CMLD013075 and geldanamycin in lysate prepared from quiescent 3T3 cells. Affinity for both CMLD013075 and geldanamycin was indeed lower in 3T3 lysate

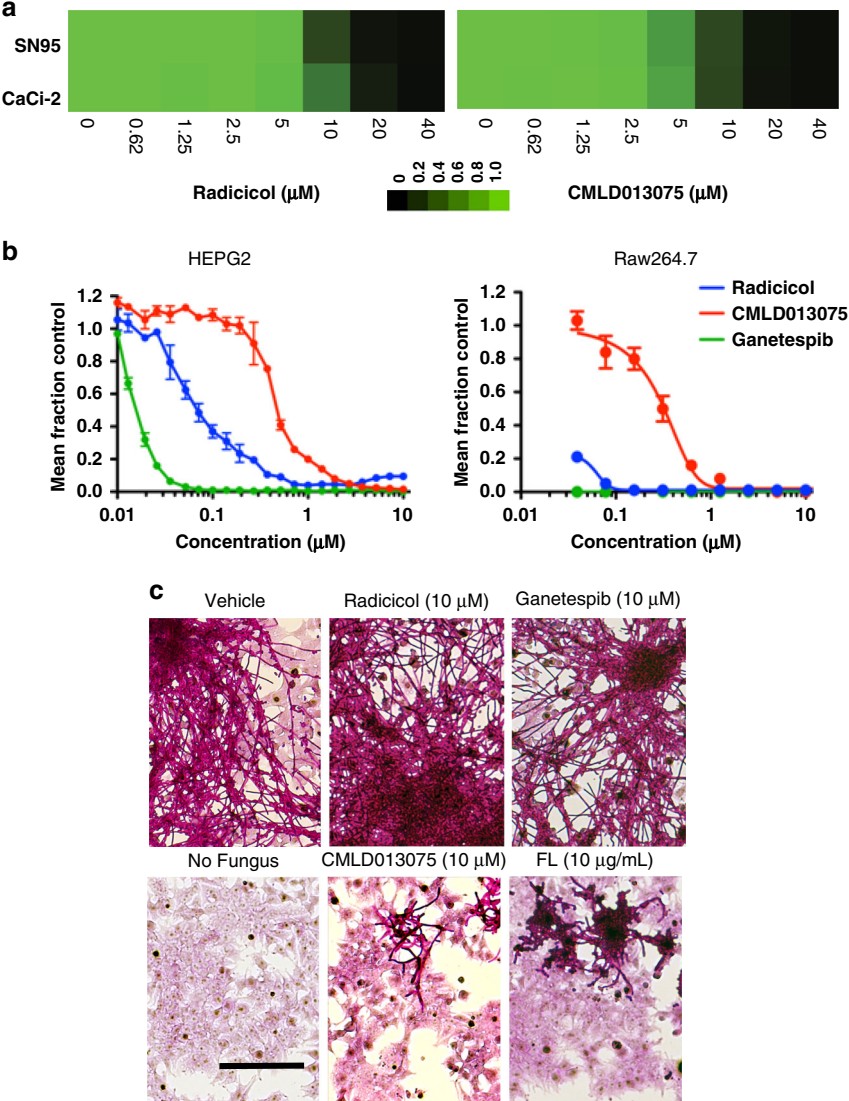

**Fig. 6** Species-selective antifungal activity of Hsp90 inhibitor CMLD013075. **a** Relative growth inhibition by Hsp90 inhibitors of reference *C. albicans* strain (SN95) and a clinical isolate (CaCi-2) with resistance to FL. The effect of 48-hour exposure to each inhibitor over a twofold dilution series of concentrations is displayed in heat-map format. Each colored box represents the mean of duplicate determinations. Color scale for relative growth is provided to far right of panel, green: no inhibition to black: complete inhibition. The experiment was repeated as an independent biological replicate to confirm results. **b** Concentration-dependent inhibition of mammalian cell growth and survival after 3-day culture with Hsp90 inhibitors. Data points depict the mean of triplicate determinations, error bars, SD. The experiment was repeated as an independent biological replicate to confirm results. **c** Monolayer cultures of human HEPG2 cells in 6-well format were infected with *C. albicans* and then incubated with the indicated compounds for 48 h (FL, conventional antifungal fluconazole). Cultures were then fixed and stained by Periodic Acid-Schiff (PAS) technique to visualize fungal burden (dark red/purple signal) and underlying human cells (light pink signal). Representative fields from conventional light photomicrographs are presented at the same magnification in all panels. The experiment was repeated in two independent biological replicates to confirm results. Scale bar, 100 µm

than HEPG2 lysate suggesting a relative improvement in selectivity for fungal Hsp90 (Supplementary Figure 7b). In light of the effects on geldanamycin affinity, however, care must be taken in the interpretation of these results because the probe used for these FP assays is itself geldanamycin-based. Although we cannot resolve the potential contributions of higher Hsp90 affinity state, higher Hsp90-dependence for proliferation/survival or even increased Grp94 content to the increased sensitivity of HEPG2 cells to CMLD013075, it is likely that using cancer cells as a comparator underestimates the compound's fungal selectivity.

To determine whether CMLD013075 retained selective activity against *C. albicans* in the presence of human cells, we optimized conditions for growth of both cell types together in a co-culture model. Established monolayers of adherent human HEPG2 cells

were infected with CaCi-2 at a multiplicity of infection (MOI) of ~0.2 and test compounds added. After 24 h of growth at 37 °C, cultures were formalin-fixed and stained to visualize fungal burden and underlying human cells (Fig. 6c). CMLD013075 (10 µM) was clearly more effective at limiting fungal overgrowth than radicicol or ganetespib. Its inhibitory effect was comparable to that of a high concentration of the widely used triazole antifungal drug fluconazole (FL) against this moderately drug-resistant isolate.

**Selectivity reduces target-related mammalian toxicity.** Compromise of host immune function and other target-related toxicities preclude the use of non-species-selective Hsp90 inhibitors

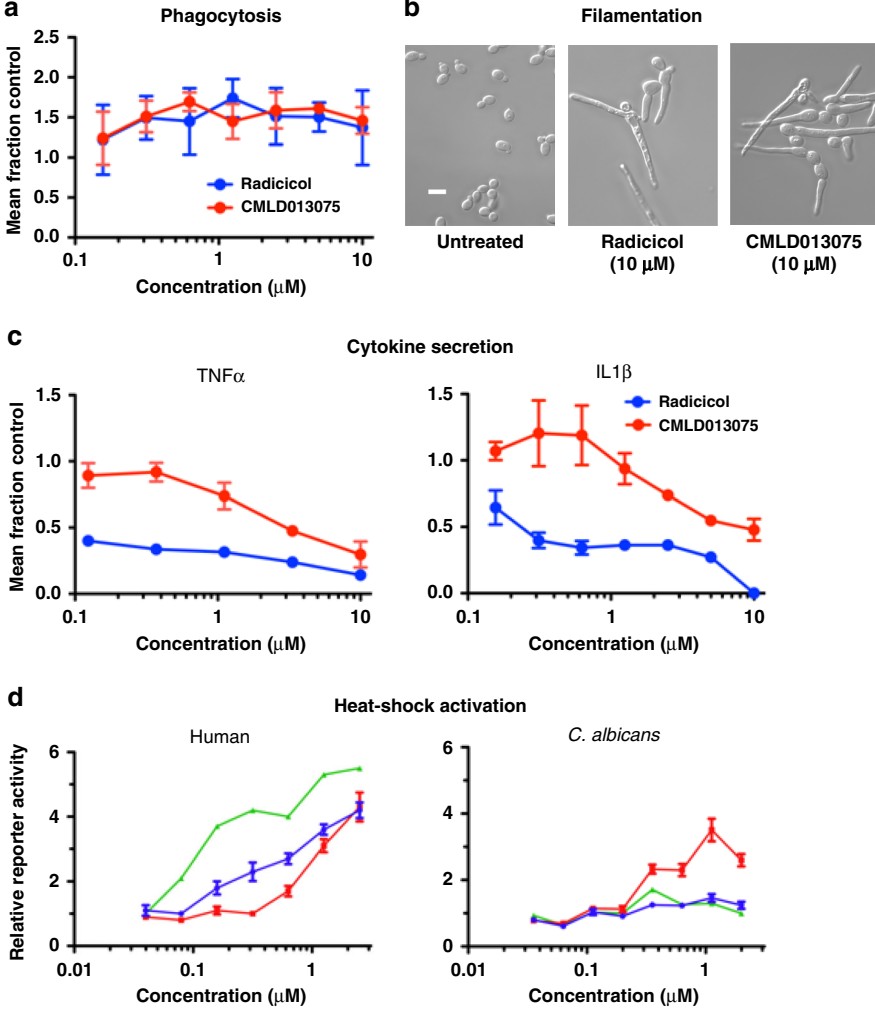

**Fig. 7** Sparing of adverse effects on mammalian cells by CMLD013075. **a** Cultures of a mouse macrophage cell line were pre-incubated with the indicated concentrations of Hsp90 inhibitors, washed into compound-free medium and infected with a strain of GFP-expressing *C. albicans*. After an additional 2-h incubation, engulfment of fungus was monitored by flow cytometry. The percentage of macrophages with internalized *C. albicans* is plotted relative to no-compound control for increasing concentrations of inhibitor. Each data point represents the mean of triplicate determinations. Error bars, SD. The experiment was repeated as an independent biological replicate to confirm results. **b** Effect of Hsp90 inhibitor exposure on fungal filamentation, a well-established consequence of Hsp90 inhibition was imaged by differential interference contrast microscopy. Representative fields from micrographs obtained at the same magnification for all images are presented. Scale bar, 10 μm. The experiment was repeated as an independent biological replicate to confirm results. **c** Cultures of a mouse macrophage cell line were pre-incubated with the indicated concentrations of Hsp90 inhibitors, washed into compound-free medium and infected with a strain of GFP-expressing *C. albicans*. After additional 2-h incubation, aliquots of medium were removed and assayed by ELISA for level of the indicated pro-inflammatory cytokines. Each data point depicts the mean derived from three independent cultures. Error bars, SD. The entire experiment was repeated as an independent biological replicate to confirm results. **d** Induction of the heat-shock response was measured in a strain of *C. albicans* (right) and a mammalian cell line (left). Cells of each species were stably transduced with a plasmid encoding a reporter protein under transcriptional control of heat-shock promoter elements from an *HSP70* gene native to the respective species. Relative concentration-dependent activation of reporter expression by compounds after overnight (human cells) or 6-h (*C. albicans*) exposure is plotted. Each point depicts the mean of three independent wells. Error bars, SD. The experiment in both human and fungal cells was repeated as an independent biological replicate to confirm results

in the treatment of fungal infections. Host defense against invasive infection relies heavily on the ingestion and destruction of *C. albicans* by innate immune cells, such as macrophages[46]. Thus, we compared the effect of CMLD013075 to that of radicicol on phagocytosis of *C. albicans* by macrophages. We pre-treated Raw264.7 mouse cells for two hours with radicicol or CMLD013075 then infected cultures with log-phase *C. albicans*. After an additional two hours, phagocytosis of the fluorescently labeled fungi was measured by flow cytometry. Neither compound impaired engulfment of *C. albicans* by these macrophages (Fig. 7a). Likewise, both compounds had similar effects in driving

the yeast-to-filament transition reported for classical Hsp90 inhibitors (Fig. 7b). This Hsp90-dependent morphogenetic program has an important role in the organism's virulence and interactions with host immune effectors such as macrophages[12].

In contrast to phagocytosis, Hsp90 inhibitor treatment did impair macrophage activation as measured by secretion of the pro-inflammatory cytokines TNF-α and IL-1β. Consistent with its selectivity for fungal Hsp90, however, CMLD013075 caused far less suppression than radicicol of cytokine production (Fig. 7c), a critical requirement for induction of robust immune responses that aid in the clearing of fungal infections. In addition to effects

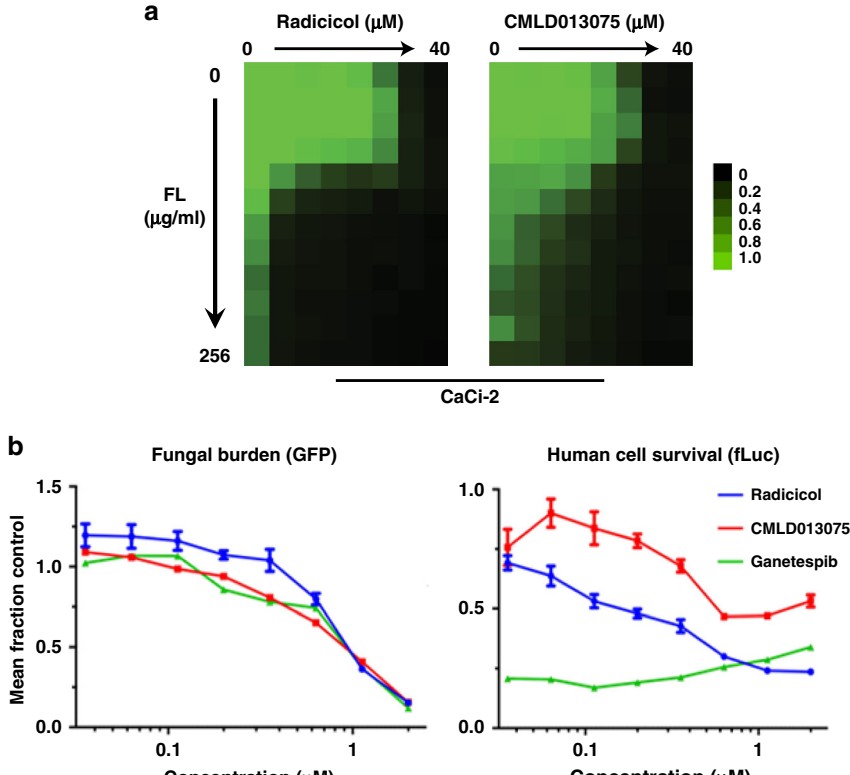

**Fig. 8** CMLD013075 reverses antifungal resistance in co-culture. **a** Checkerboard analysis of antifungal activity for combination of FL with Hsp90 inhibitors. FL-resistant strain CaCi-2 was exposed to the indicated twofold serial dilutions of each compound for 48 hours. Relative growth inhibition is displayed in heat-map format. Each colored box represents the mean of duplicate determinations. Color scale for relative growth is provided to the right of panel, green: no inhibition to black: complete inhibition. The experiment was repeated as an independent biological replicate to confirm results. **b** Co-culture of fluconazole (FL)-resistant, GFP-expressing *C. albicans* strain with human 293T cells line stably expressing a luciferase reporter. The concentration-dependent inhibition of fungal (left) and human (right) cell growth and survival by Hsp90 inhibitors in the presence of FL was measured in 384-well format after 48-h treatment. Relative fungal burden was evaluated by plate reader using a fluorescence endpoint while human cell survival was evaluated by measuring luminescence. Values for each endpoint were normalized to wells receiving no Hsp90 inhibitor. Each point represents the mean of measurements from triplicate wells. Error bars, SD. The experiment was repeated once as an independent biological replicate to confirm results

on host immune function, we also examined activation of the evolutionarily conserved heat-shock response by CMLD013075. Such activation is well-defined toxicity of impairing Hsp90's essential role in maintaining protein homeostasis in cells and has been widely used as a sensitive pharmacodynamic endpoint for target engagement in the clinical development of Hsp90 inhibitors as anticancer agents[47,48]. To compare the degree to which CMLD013075 vs classical Hsp90 inhibitors activated the heat-shock response, we used cells stably expressing reporter proteins (β-galactosidase in *C. albicans* and firefly luciferase in human 293T kidney-derived cells) under transcriptional control of heat-shock promoter elements from an *HSP70*-family gene of the relevant species. Treatment of human cells with the anticancer Hsp90 inhibitor ganetespib or radicicol itself drove concentration-dependent reporter activation as expected. Although CMLD013075 also activated the reporter, activation required much higher concentrations of this compound (Fig. 7d). The opposite pattern of potency held true in *C. albicans*; CMLD013075 induced a peak threefold activation of the heat-shock reporter at 1 μM whereas neither ganetespib nor radicicol induced the reporter to a measurable extent (Fig. 7d).

**Reversal of antifungal drug resistance.** To determine whether CMLD013075 could potentiate antifungal activity in a manner similar to that previously reported for other Hsp90 inhibitors[6,7],

we used a standard dose-response matrix (checkerboard) approach involving gradients of Hsp90 inhibitor and FL, the most commonly used clinical antifungal. By Bliss independence modeling analysis[49], we found strong synergy for both radicicol ($\delta =$ 22.6) and CMLD013075 ($\delta = 13.0$) in combination with FL against the *C. albicans* clinical isolate CaCi-2 (Fig. 8a). Given that azoles such as FL are fungistatic against *C. albicans* alone but fungicidal in combination with an Hsp90 inhibitor[6,7], we also performed clonogenic survival assays with CMLD013075 to determine whether it conserved this important property of Hsp90 inhibition. Using a reference strain of *C. albicans*, we again performed checkerboard assays titrating FL and CMLD013075. After 48 h in liquid culture (Supplementary Figure 9a), cells were transferred onto drug-free solid medium to assess viability (Supplementary Figure 9b). With similar potency, both radicicol and CMLD013075 transformed the fungistatic azole into a fungicidal combination.

To learn whether CMLD013075 would eliminate the FL-resistant growth of strain CaCi-2 in a manner similar to that previously reported for other Hsp90 inhibitors, the wells in plates that had been previously seeded with luciferized human cells were infected with GFP-marked CaCi-2. Hsp90 inhibitors were then added over a range of concentrations in addition to a fixed concentration of FL. After 48 h at 37 °C, relative fungal burden was measured by GFP fluorescence and survival/proliferation of the human cells was evaluated by luminescence. While all Hsp90

inhibitors displayed comparable activity in sensitizing CaCi-2 to FL, CMLD013075 was much less toxic than radicicol or ganetespib to HEPG2 cells in the co-culture environment (Fig. 8b).

Encouraged by the selective activity of CMLD013075 in co-culture, we made preliminary investigations into the compound's suitability for administration to mice. To do so, we first developed a sensitive and reliable analytical method to measure levels of the compound in biological media (Supplementary Figure 10a). We then measured the stability of the compound in mouse plasma. As expected, by dampening its Michael reactivity, the oxime modification in CMLD013075 increased plasma stability compared to radicicol (Supplementary Figure 10b). Unfortunately, the pendant 4-methoxybenzyl group also increased susceptibility to P450 metabolism, as reflected by ~80% loss of parent radicicol oxime after 60-min incubation in mouse liver microsomes. Given this pharmacological liability, relatively rapid clearance from the systemic circulation in a single dose pharmacokinetic study of CMLD013075 in mice was observed (Supplementary Figure 10c). Further work will be required to improve the whole-animal pharmacology of our tool compound CMLD013075 before testing in mice can be pursued. Nevertheless, results so far provide strong structural evidence and biological proof-of-principle for the feasibility of targeting Hsp90 in a species-selective manner to combat invasive fungal disease.

## Discussion

Heat-shock proteins, especially Hsp90, are notable for their exceptional degree of sequence conservation across species. Despite such conservation, the crystal structures of the C. albicans Hsp90 NBD presented here revealed sufficient conformational flexibility to enable synthesis of first a fungal-selective Hsp90 inhibitor. Our proof-of-concept probe molecule CMLD013075 inhibited the growth of C. albicans, enforced filamentous growth in the absence of an inducing cue, displayed reduced toxicity against mammalian cell lines and potentiated azole activity against a drug-resistant clinical isolate of C. albicans.

In initial work, the crystal structures of apo, ADP-bound, and radicicol-bound fungal NBD showed minimal structural variation. More problematic, they also showed little variation with respect to the human NBD. Importantly, however, the region spanning residues Asp91 to Gly97 of C. albicans NBD in its apo state was seen in two conformations. We hypothesized that this observation might result from greater flexibility within the fungal ATP-binding site than was immediately obvious, an increased flexibility that might contribute to observed differences in the intrinsic ATPase activity of the fungal and host homologs[24,50]. The significance of this observation, however, became clearer when we determined structures for the C. albicans NBD in complex with SNX-2112 and AUY922. The region spanning residues 93–102 adopted very distinct configurations to accommodate each of these compounds. In addition, extensive structural rearrangements, inclusive of regions outside of the nucleotide-binding site were observed in the case of AUY922. In complex with this drug, movement of the NBD lid region was accompanied by disappearance of the N-terminal β-strand. Remarkably, previous work in S. cerevisiae has shown that the first 24 residues of the NBD stabilize it in a rigid state and their removal confers flexibility specifically to the region between amino acids 98 and 120[51]. While no such data are available in Candida, our findings are consistent with N-terminal alterations being capable of affecting the architecture of the overall binding site and vice versa.

Insights gained from the fungal structures reported here inspired synthesis of derivatives of the natural product radicicol

and its des-chloro variant monocillin I, all synthesized with the goal of achieving fungal selectivity. We hypothesized that steric challenges would allow for selective accommodation by the more flexible fungal NBD. Amongst the analogs, CMLD013075 showed >25-fold biochemical selectivity for C. albicans Hsp90 NBD compared with the mammalian homolog. Importantly, a co-crystal structure for CMLD013075 in complex with the fungal NBD also showed a major and distinct structural change associated with inhibitor binding. These changes could be due to either stabilization of a pre-existing conformation visited by the NBD during the Hsp90 chaperoning cycle or de novo induced-fit effects, but both mechanisms are dependent on the intrinsic protein flexibility[52]. In the case of CMLD013075, the orientation of the p-methoxybenzyl group towards the protein's core is proposed to support these conformational changes, with stabilizing hydrophobic and NH⋯π interactions in a newly formed elongated pocket. This hypothesis is supported by our mutagenesis results demonstrating that the mutations Leu130Ala (affecting ligand-binding) and Phe131Tyr (affecting lid region flexibility) were accompanied by a reduction in the binding affinity of CMLD013075. Notably, these positions are variable in other fungal pathogens, including C. neoformans and A. fumigatus. Such variation could negatively impact CMLD013075 binding affinity in these species and necessitate additional modifications in the pursuit of a pan-selective fungal Hsp90 inhibitor.

Given the remarkable conformational shift associated with the binding of monocillin I-derived CMLD013075, it remains to be determined whether radicicol-derived analogs (3–5), which exhibited more modest fungal selectivity and similar whole-cell bioactivity to CMLD013075, bind in a similar manner. The sole difference between radicicol and monocillin I is a single chlorination of the aryl ring. Among resorcylate Hsp90 inhibitors, replacement of this chlorine with moderately sized aliphatic groups (ethyl, isopropyl) often improves potency, attributed to enhanced hydrophobic interactions in the lipophilic pocket of the ATP-binding region, as seen in the C. albicans and human NBD-AUY922 co-crystals[37]. The top fungal-selective radicicol-derived inhibitor, ortho-chlorobenzyl-substituted 4 (2.9-fold selectivity in lysates, 531 nM EC$_{50}$, 2.6 μM MIC), would be unable to engage in hydrophobic interactions as observed at the para-position of the CMLD013075 benzyl oxime, and halogenation of this ring should also significantly decrease potential for NH⋯π binding with Asn40.

Hsp90 orthologue selectivity has been recognized as a promising toxicity-mitigating strategy to improve the clinical efficacy of Hsp90 inhibitors in humans[53]. Several distinct chemotypes have been shown to take advantage of the differential structural plasticity intrinsic to each human Hsp90 orthologue, thus bypassing the limited sequence variation seen in these essential chaperones[54–56]. The resorcylate analog CMLD013075 achieves fungal specificity in an analogous manner, exploiting the extensive structural flexibility of the C. albicans NBD. Nonetheless, there are significant variations in the structural repertoire available to the fungal and host chaperones. Both proteins can accommodate compounds such as SNX-2112 with a limited remodeling of the binding site. Only the fungal protein, however, showed remodeling of the N-terminus and lid region to accommodate the binding of CMLD013075 and AUY922. The increased catalytic efficiency of the fungal chaperone compared with the human protein might well be correlated with its enhanced flexibility, but from the perspective of drug discovery, such conformational flexibility also allows access to unique structural features to generate fungal-selective compounds.

The observed conformational change in the fungal NBD showed some similarity to the open conformation described for Grp94 in complex with ADP (r.m.s.d. of 4.337 Å for main-chain

atoms), and it can be related to an initial conformational change necessary to accomplish a complete catalytic cycle by the full-length chaperone[57]. To investigate orthologue selectivity further, we also overlaid the structure of our *C. albicans*-CMLD013075 complex with the structure reported for Grp94 in complex with PUWS13, a paralogue-selective inhibitor of this chaperone (PDB code 3O2F)[58]. Consistent with the 5-fold lower affinity of CMLD013075 for Grp94 NBD than fungal Hsp90 NBD, however, these two complexes shared very limited structural similarity (r. m.s.d. of 8.39 Å for main-chain atoms).

Over the past several decades, an extensive literature has accumulated characterizing the role of the molecular chaperone Hsp90 in diverse human diseases including cancer[23], neurode-generative disorders[27], and the establishment and maintenance of infectious diseases[32,59]. In concert, tremendous efforts have been made to develop potent and specific drugs with which to target this chaperone in humans. In oncology, where most if not all clinical trials have taken place, these compounds have displayed disappointing activity[60]. However, based on findings in model systems ranging from cancer cells to fungi, we suggest the true therapeutic potential of Hsp90 inhibitors actually lies in combining them with other therapies to alter evolvability and limit the emergence of drug resistance. The discovery and development of species-selective Hsp90 inhibitors capable of potentiating current antifungals in this unique way could make it possible to test this fundamental hypothesis in patients and help combat the growing threat of invasive fungal infection. To provide effective therapeutics, however, the challenge of designing inhibitors that are not only fungal-selective, but also able to access the intracellular environment and avoid efflux will need to be overcome. Pathogenic fungi have evolved an array of permeability constraints for antifungal drugs, via compromised drug import and/or increased drug efflux[61,62]. The importance of these constraints is highlighted by the disparity between the low nanomolar affinity of CMLD013075 we demonstrated for Hsp90, and the micromolar concentrations needed to inhibit whole-organism growth and survival in culture. Hopefully, despite such pharmacological challenges, the cross-over approach we adopted in leveraging extensive anticancer drug discovery/development resources to advance far less well-funded antimicrobial efforts will encourage similar efforts on the part of other investigators. Finally, beyond the specific application of targeting Hsp90, ou\r findings highlight the value of chemo-structural approaches in overcoming the broader challenge of selectively targeting members of closely conserved protein families with high biomedical significance.

## Methods

**Reagents**. The Hsp90 inhibitors SNX-2112 (S2639, >99%) AUY922 (S1069, 99%) ganetespib (S1159, >99%) were purchased from Selleck Chemicals. Geldanamycin (ant-gl-5, >95%) was purchased from Invivogen. Fluconazole was obtained from Sequoia Research Products (SRP01025f, >99%) and Cy3B-Mono NHS ester was purchased from GE Healthcare UK Ltd (PA63101, >95%). The cell lines HEPG2 (HB-8065), Raw264.5 (TIB-71), 293T (CRL-3216), and 3T3 (NIH/3T3; CRL-1658) were obtained from the American Type Culture Collection (ATCC) and grown in DMEM supplemented with 10% fetal bovine serum at 37 °C under 5% $CO_2$. Cultures were confirmed negative for mycoplasma contamination by monthly surveillance testing using a PCR-based kit.

**Chemical synthesis**. See Supplementary Methods for materials, methods, procedures and characterization data for the synthesis of oxime analogs 1–8.

Cy3-labeled geldanamycin (Cy3-GdA) was prepared by reacting geldanamycin with a 2-fold molar excess of diaminobutane in dimethylformamide[34]. After clean-up by silica gel chromatography, amine-derivatized geldanamycin was reacted with Cy3B-Mono NHS ester and the conjugate purified by silica gel chromatography. Aliquots were dried down and stored in the dark under argon prior to use in FP assays.

**NBD expression and purification for crystallographic studies**. Sequences of the recombinant NBD used for crystallization and FP binding assays were as follows:

*C. albicans* Hsp90 (E7-K218)

mhhhhhhssgrenlyfqgETHEFTAEISQLMSLIINTVYSNKEIFLRELISNASDALD KIRYQALSDPSQLESEPELFIRIIPQKDQKVLEIRDSGIGMTKADLVNNLGTIAK SGTKSFMEALSAGADVSMIGQFGVGFYSLFLVADHVQVISKHNDDEQYVWES NAGGKFTVTLDETNERLGRGTMLRLFLKEDQLEYLEEKRIKEVVKKHSEFVAY PIQLVVTKEVEK Human Hsp90 beta (M1-K223) mhhhhhhssgrenlyfqgMPEEVH HGEEEVETFAFQAEIAQLMSLIINTFYSNKEIFLRELISNASDALDKIRYESLTDP SKLDSGKELKIDIIPNPQERTLTLVDTGIGMTKADLINNLGTIAKSGTKAFMEA LQAGADISMIGQFGVGFYSAYLVAEKVVVITKHNDDEQYAWESSAGGSFTV RADHGEPIGRGTKVILHLKEDQTEYLEERRVKEVVKKHSQFIGYPITLYLEKE REK

Human Grp94 (A63-I337)

maswhpqfekgaddddkmASQIRELREKSEKFAFQAEVNRMMKLIINSLYKNKEIFL RELISNASDALDKIRLISLTDENALSGNEELTVKIKCDKEKNLLHVTDTGVGM TREELVKNLGTIAKSGTSEFLNKMTEAQEDGQSTSELIGQFGVGFYSAFLVAD KVIVTSKHNNDTQHIWESDSNEFSVIADPRGNTLGRGTTITLVLKEEASDYLE LDTIKNLVKKYSQFINFPIYVWSSKTETVEEPMEEEEAAKEEKEESDDEAAVEE EEEEKKPKTKKVEKTVWDWELMNlpfssisahhhhhhhhhh

Recombinant NBDs were produced on small or large-scale using different methods. For small scale, the protein expression construct was transformed into *E. coli* Rosetta2 (Novagen) competent cells. Successful transformants were grown in LB overnight at 37 °C, these cultures were used to inoculate 1 L Terrific Broth (TB) cultures. Growth at 37 °C was continued until $OD_{600}$ reached ~1.0, whereupon temperature was lowered to 18 °C and protein expression was induced by the addition of 1 mM isopropyl β-D-thiogalactopyranoside (IPTG). Overnight post-induction cells were collected by centrifugation and resuspended in binding buffer (50 mM Hepes pH 7.5, 500 mM NaCl, 5 mM imidazole, 5% glycerol, 1 mM benzamidine, and 1 mM phenylmethanesulfonyl fluoride (PMSF)) and the resuspended cells were stored at −80 °C. For large-scale production, transformants were grown in TB media in a LEX bioreactor system (Harbinger Biotechnology and Engineering Corp., Ontario, Canada). Overnight starter cultures were allowed to grow at 37° C until reaching an $OD_{600}$ value ~5, cooled to 15 °C, and subsequently induced overnight with 0.5 mM IPTG. Cells were harvested by centrifugation and cell pellets resuspended in 40 mL of binding buffer per liter of culture, then flash frozen in liquid nitrogen and stored in −80 °C until needed.

Frozen cells were thawed in 0.5% CHAPS and 500U of benzonase for ~45 min at room temperature. For large-scale preparations, an additional mechanical lysis was done using an M-110EH Microfluidizer Processor (Microfluidics Corp., MA, USA). Cell lysates were then centrifuged to eliminate debris and the cleared lysate was loaded onto a DE52 anion exchange resin column (Whatman, MA, USA) followed by a 2 mL Ni-NTA column (Qiagen, MD, USA). This column was then washed with 200 mL of a buffer consisting of 50 mM HEPES, pH 7.5 with 500 mM NaCl, 30 mM imidazole, and 5% glycerol. The protein was eluted with 7–15 mL of a buffer consisting of 50 mM HEPES, pH 7.5 with 500 mM NaCl, 250 mM imidazole and 5% glycerol. Following elution, 1 mM EDTA and 1 mM TCEP were added to the sample. The protein was further purified by size exclusion chromatography on a Superdex 200 (GE Healthcare, NJ, USA) column equilibrated with buffer consisting of 50 mM HEPES, pH 7.5 and 500 mM NaCl. The protein was concentrated in an Amicon Ultra centrifugal filter device (Millipore, MA, USA) and its identity was confirmed by SDS-PAGE and mass spectroscopy.

**Protein crystallization and structure determination**. Aliquots of purified recombinant *C. albicans* Hsp90 N-terminal domain (NTD) were crystallized at 21 °C using a sitting-drop vapor-diffusion method, either alone (*apo*) or in the presence of five different ligands, ADP, radicicol, CMLD013075, SNX-2112 and AUY922. Crystals were obtained by mixing one part of protein solution at 10–15 mg/mL (10 mM HEPES, pH 7.5 with 500 mM NaCl, 2 mM TCEP and 4 mM $MgCl_2$) with or without 2 mM ligand with one part of reservoir solution. In the case of *apo* and ADP co-crystals, the reservoir solution contained 2 M ammonium sulfate, 2% PEG400, 100 mM HEPES, pH 7.5. For radicicol co-crystals, the reservoir solution contained 8% Tacsimate, pH 5.0 and 20% PEG3350; for CMLD013075 it included, 5 mM $CoCl_2$, 12% PEG3350, 100 mM HEPES, pH 7.5; for the SNX-2112 complex crystals it contained, 200 mM $MgCl_2$, 100 mM bis-Tris, pH 6.5, and 25% PEG3350. Finally, for the AUY922 complex crystals it contained, 20 mM $MgCl_2$, 100 mM HEPES, pH 7.5, and 22% polyacrylic acid sodium salt. Crystals appeared within one week and were cryo-protected with 20–25% ethylene glycol or glycerol supplemented mother liquor before being flash cooled in liquid nitrogen.

Diffraction data were collected using home X-ray sources for the *apo* and ADP structures. A synchrotron X-ray (beamline 08ID-1 at the Canadian Light Source (CLS)) was used for radicicol and CMLD013075 complexes. Beamline 23ID-D at APS - Argonne National Laboratory was used for SNX-2112 and AUY922 complexes. Data were processed with HKL3000[63,64]. Structures were determined by molecular replacement using the yeast Hsp90 NBD (PDB code 1AH8)[65] as a search model for the first complex structure of the *C. albicans* Hsp90 NBD. This structure was used as the model for subsequent structures. Phaser[66] was used for molecular replacement calculations, while model building was performed with COOT[67] and structures were refined with Phenix[68]. The radicicol and SNX-2112 complex crystals were twinned, and xtriage (as implemented in Phenix)[69] identified the following merohedral twin law *h,-k,-l* and a twin fraction of ~ 0.3 for both crystals. The positions of all compounds in the reported *C. albicans*

Hsp90-ligand complex crystal structures were unambiguous as indicated by their simulated annealing OMIT maps (Figure S11). CMLD013075 coordinates and geometry restraints were created within the ELBOW. The stereochemistry of both models was checked by MOLPROBITY[70]. Relevant data collection and refinements statistics are shown in Table S1. Protein structure superpositions were calculated with LSQKAB[71] as implemented in CCP4[72]. Protein structure figures were generated using UCSF Chimera package[73]. C. albicans and human Hsp90-AUY922 2D interaction diagrams (Figure S4b bottom panels) were generated using Maestro (Schrödinger Release 2016–1: Maestro, Schrödinger, LLC, New York, NY, 2016).

**Yeast strains and culture conditions**. Archives of all yeast strains were maintained at −80 °C in 25% glycerol. Strains used in experiments were maintained on solid (2% agar) yeast extract peptone (YPD, 1% yeast extract, 2% bactopeptone, 2% glucose) at 4 °C for no more than one month. For experiments, yeast strains were routinely grown in either YPD medium, unless otherwise indicated. Strains used in this study are listed in Supplementary Table 4. Strain construction is described in Supplementary Methods.

**Plasmid construction**. Recombinant DNA procedures were performed according to standard protocols and construction of vectors is described in Supplementary Methods Supporting Information. Plasmids were sequenced to verify the absence of any nonsynonymous mutations. Plasmids used in this study are listed in Supplementary Table 5. Oligonucleotides used in this study are listed in Supplementary Table 6.

**Minimum inhibitory concentration assays**. Antifungal tolerance and resistance were determined in flat bottom, 96-well microtiter plates using a modified broth microdilution protocol as described[8,10]. All Hsp90 inhibitors were formulated in dimethyl sulfoxide (DMSO, Sigma Aldrich Co.); fluconazole was dissolved in sterile ddH$_2$O. Each strain was tested in duplicate on at least two occasions. MIC data were quantitatively displayed in heat-map format using the program Java TreeView 1.1.3 (http://jtreeview.sourceforge.net). To test for fungicidal activity, cultures from MIC plates were spotted on YPD agar plates using a spotter (Frogger, V&P Scientific, Inc). Plates were photographed after 48 h of incubations at 30 °C.

**Preparation of whole-cell lysates for FP assays**. To prepare human cell lysate, approximately $1 \times 10^7$ HEPG2 or 3T3 cells were collected by centrifugation at 4 °C, washed in cold PBS and resuspended in 250 μL binding buffer (20 mM HEPES pH 7.5, 50 mM KCl, 5 mM MgCl$_2$, 0.01% Triton X-100) supplemented fresh with a protease inhibitor cocktail tablet (Roche; 1183617001), phosphatase inhibitor tablet (Roche; 04906837001), 20 mM Na$_2$MoO$_4$, and 1 mM DTT. After vortexing, cell lysate was frozen at −80 °C for 2 h, followed by thawing on ice. The vortex-freeze-thaw step was repeated. Complete lysis was confirmed by microscopy followed by centrifugation at 14,000 rpm for 30 min at 4 °C. To prepare yeast lysates, an overnight culture grown in YPD was diluted to an OD$_{600}$ of 0.15 in 1 L of pre-warmed YPD. This culture was grown for 4 h at 37 °C, then centrifuged at 4000 rpm for 20 min at 4 °C. The pellet was resuspended in 10 mL of the same binding buffer used to prepare mammalian lysate, passed twice through a French Press and then centrifuged at 14,000 rpm for 30 min at 4 °C. Post centrifugation, yeast and mammalian lysates were filtered (0.22 μm), supplemented with sterile glycerol to a final concentration of 15% (v/v), flash frozen, and stored at −80 °C for up to 5 months. Aliquots were thawed immediately prior to use.

**FP assays**. Total protein concentration of human and yeast lysates was determined by Bradford assay as described previously[8]. Titrations of Cy3-GdA probe and lysate were evaluated to define conditions that resulted in 75% maximal probe polarization with no competitor present. Serial dilutions of test Hsp90 inhibitors were then assayed under these same conditions to monitor loss of fluorescence polarization as an indicator of probe displacement from Hsp90. All determinations were performed in duplicate wells using 384-well black flat-bottom microtiter plates (Greiner Bio-One; 655076). Titrations of test compound in 25 μL of binding buffer (supplemented with 0.1 mg/mL bovine gamma globulin), were mixed with an equal volume of freshly prepared whole-cell lysate spiked with Cy3-GdA (0.1 nM). Plates were incubated at room temperature for 4–5 h to achieve equilibrium binding for the geldanamycin-based probe[35]. Signal in millipolarization (mP) units was measured at an excitation wavelength of 535 nm and emission wavelength of 595 nm in a SpectraMax i3 microplate reader (Molecular Devices) using Softmax Pro software (version 5.4.1). Each experiment was repeated in at least biological duplicate. Non-linear 4-parameter curve fitting of raw displacement data was performed in GraphPad Prism 5.0 to determine EC$_{50}$ values as a measure of relative Hsp90-binding affinity. Results were normalized to the value determined for GdA in lysate of each cell type. To determine assay-independent inhibitory K$_i$ values for compounds by FP, the binding affinity of Cy3-GdA probe (0.1 nM) for each NBD (human and yeast Hsp90 10 nM; human Grp94 7.5 nM) was first determined by saturation binding measurements[33]. Using these values, data from equilibrium competition binding experiments were then analyzed by methods that take into account the non-ligand saturated conditions inherent to FP assays to calculate the K$_i$ of inhibitors[33].

**Microscopy**. C. albicans strain SN95 was sub-cultured to an OD$_{600}$ of 0.1 and grown for 6 h in the absence or presence of Hsp90 inhibitors as indicated. Imaging of cells cultured in liquid was performed by differential interference contrast microscopy with a Zeiss Axio Imager.MI and Axiovision software (Carl Zeiss, Inc.).

**Mammalian cell toxicity**. Cancer cells were plated in 96-well format at a density of 5000 per well (HEPG2) or 2000 per well (Raw264.7) in DMEM supplemented with 10% fetal bovine serum (FBS) and 1× Pen-Strep antibiotic. Non-tumorigenic mouse 3T3 fibroblasts were plated in 384-well format at 2500 per well. After overnight incubation, two-fold dilutions of test compounds were added to wells for 72 h followed by measurement of relative viable cell number using standard resazurin dye-reduction assay and microplate fluorometer (Tecan Infinite M1000Pro, 544 nm excitation and 590 nm emission) as previously published[74].

**Phagocytosis assay**. The ingestion of GFP-marked C. albicans CaCi-2 (CaLC867, Table S4) by mouse macrophages was monitored using flow cytometry. Raw264.7 macrophage cultures were pre-treated with various concentrations of Hsp90 inhibitors for 2 h at 37 °C and then infected with fungus at a multiplicity of infection of ~2 for an additional 2 h. Wells were harvested by trypsinization and resultant cell suspensions incubated with Guava ViaCount reagent per manufacturer's recommendation (Millipore Sigma, Cat #4000-0040). Red fluorescent signal was used to identify macrophages while green fluorescent signal identified the fungus. Dual positive (red and green) events with appropriate forward- and side-scatter parameters were quantitated as representative of ingestion of fungus by a macrophage. All determinations were performed in triplicate and the experimental design was repeated twice.

**Cytokine production**. Raw264.7 cells were plated in 100 μL of media (DMEM supplemented with 10% fetal bovine serum (FBS) and 1X Pen-Strep antibiotic) at a density of $1.5 \times 10^5$ cells per well in a 96-well plate. The next day, cells were treated with a dilution series of radicicol or CMLD013075. After 2 h incubation at 37 °C, medium was flipped off and wells washed two times with 200 μL of warm medium. During incubation of macrophages with Hsp90 inhibitor, a culture of C. albicans grown in YPD overnight at 30 °C was spun down and washed twice with PBS. Cell counts were then determined using a hemocytometer and an inoculum of $6 \times 10^5$ cells per well was added to the macrophages. Co-culture incubation time was 2 h for measuring secretion of TNF-α and 5 h for measuring IL-1β cytokine levels. To monitor IL-1β production, macrophages were primed with 50 ng/mL of LPS (Sigma; L2630) for 2 h prior to infection with C. albicans. At the end of each experiment 150 mL of media was removed and added to a new 96-well plate that contained a 25 μL of solution of 7× protease inhibitors in water. The plate was frozen and stored at −80 °C until measurement of cytokine levels using ELISA kits from R&D Systems: TNF-α DuoSet (DY410) and IL1-1β/IL-1F2 DuoSet (DY401).

**Heat-shock induction assays**. Concentration-dependent activation of the heat-shock response in human cells by Hsp90 inhibitors was measured in 96-well format as previously published using 293T cells stably transduced with a reporter construct encoding GFP-luciferase fusion protein under transcriptional control of heat-shock elements from the HSP70B gene[74]. Activation of the response in C. albicans was also measured in 96-well format using a reporter strain (CaLC922) engineered with HSP70 promoter elements driving expression of the LacZ gene from Streptococcus thermophilus integrated at the HSP70 locus with the URA3 marker for selection. After culture for 2 h in the presence of serial dilutions of Hsp90 inhibitor, the induction of galactosidase activity was measured using luminescent Gal-Screen reagent per manufacturer's recommendations (Applied Biosystems; Bedford, MA Cat#T1029).

**Co-culture experiments**. Monolayers of HEPG2 cells established in 6-well format were infected with log-phase C. albicans (CaCi-2) at a concentration of $2.5 \times 10^3$ cells/mL in DMEM supplemented with 5% fetal bovine serum and pen-strep antibiotic. After addition of test compounds, culture was continued for 48 h. Wells were fixed in 10% formalin, stained with a periodic acid-Schiff reagent kit (Sigma, 395B-1KT) per manufacturer's recommendation and photomicrographs obtained under standard white light illumination. To quantitate the concentration-dependent toxicity of Hsp90 inhibitors in human-fungal co-cultures, 293T cells stably expressing firefly luciferase were plated in 384-well format (2000 cells in 20 μL per well, black clear-bottom plates) and allowed to adhere overnight. Wells were then infected with log-phase GFP-marked CaCi-2 (CaLC867, $2.5 \times 10^3$ cells/mL) in an equal volume of medium supplemented with fluconazole (4 μg/mL). After 48 h, medium was replaced with PBS and relative fluorescence per well measured on a plate reader (Tecan Infinite M1000Pro, 480 nm excitation and 540 nm emission). Steady-Glo luciferase assay reagent (10 μL per well, Promega Cat# E2520) was then added, plates incubated at room temperature for 10 mins and relative luminescence per well measured using an Envision plate reader (Perkin Elmer).

**Pharmacological studies**. See Supplementary Methods for materials, methods and instrumentation used to evaluate the metabolic stability in vitro and single dose plasma pharmacokinetic profile in mice of CMLD013075. Pharmacokinetic studies

in mice were performed in an ethical manner under animal study protocol 0516-037-19 as reviewed and approved by the Massachusetts Institute of Technology Committee on Animal Care (CAC).

**Statistical methods**. GraphPad Prism 5 and 7.0 were used to perform curve fitting, calculate $EC_{50}$ values and generate graphical displays. Technical replicate numbers ($n$ values) for all quantitative data characterizing the biochemical and biological activities of compounds are provided in the relevant figure legends. Informative biological experiments and compound binding measurements were repeated at least once in an independent experiment to confirm results.

## Data availability

The coordinates for NBD structures and their structure factors have been deposited with the Protein Data Bank (http://www.pdb.org) under the following accession codes: 6CJI (*apo*), 6CJJ (ADP), 6CJL (radicicol), 6CJP (CMLD013075), 6CJR (SNX-2112), 6CJS (AUY922). The source data and calculations underlying Figs. 2a, b, 3b, 4c, Supplementary Figures 3a-c and 7b are provided within the "Source Data" Excel file associated with the manuscript. Other data are available from the corresponding author upon reasonable request.

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

## Acknowledgements

We thank L. Gunatilaka (University of Arizona) for production and isolation of the radicicol and monocillin I initially used in the study. We also gratefully acknowledge Novartis Pharma AG for a generous donation of radicicol to this project. We thank C. Lewis (Whitehead Institute Metabolomics Core) for assistance with bioanalytical assays. Financial support was provided by NIH grant R01AI120958-01A1 (D.J.K, L.E.B. and L.E.C.) Additional support was provided by CIHR grants MOP-86452 and FDN-154288, and by Grand Challenges Canada Star in Global Health award 0214-01 (L.E.C.). S.L. was an HHMI Investigator.

## Author contributions

L.W., S.L., L.E.B., and L.E.C. conceived and designed the overall study. C.S.N., A.H., R.H., and J.P. designed, performed and analyzed structural studies with J.P. providing overall supervision. D.S.H., R.T. J.A.P., and L.E.B. planned, synthesized, and purified resorcinylic oximes with L.E.B. providing overall supervision. L.W., N.R., C.A.M., E.V.L, T.S.-G., C.C., S.C., J.L.X., and U.T. planned, performed, and analyzed microbiological, genetic, cell culture, biochemical, and pharmacokinetic studies with L.E.C. providing overall supervision. L.W., N.R., D.J.K. L.E.B., J.P., and L.E.C. wrote and edited the manuscript.

## Additional information

**Competing interests:** L.E.C and S.L. are named as inventors on USPTO 8343913. L.W. and L.E.C. are scientific co-founders and stock holders of Bright Angel Therapeutics (BAT), a company pursuing the discovery and development of new antifungals. J.P. is a paid consultant to BAT. The remaining authors declare no competing interests.

