## [Peer Review File · Nature Communications]

Reviewers' comments:

Reviewer #1 (Remarks to the Author):

This report by Whitesell et al. provides proof-of-concept data that support the feasibility of developing a fungal-specific Hsp90 inhibitor with minimal ability to inhibit host (human) Hsp90. Highlighting the importance of this work, these and other authors have previously shown that pharmacologic inhibition of fungal Hsp90 using inhibitors developed against the human protein provides a unique means of countering fungal resistance to standard of care agents. As part of this study, the authors found that the N-domain drug binding region is more flexible compared to that of human Hsp90, suggesting more opportunities to modify existing inhibitors to take advantage of this conformational flexibility in order to design a species-specific inhibitor. The authors report identification of such an inhibitor, a derivative of the natural product Hsp90 inhibitor radicicol, with a 4.5-fold fungal binding selectivity. Interestingly, they show that the clinically evaluated Hsp90 inhibitor SNX-2112 also displays an almost 4-fold higher affinity for fungal Hsp90 than for human Hsp90, suggesting that modification of this chemotype may provide another route to develop a fungal-specific inhibitor with preferred in vivo properties. While the preliminary attempts at in vivo evaluation of CMLD013075 were disappointing due to its poor PK, etc., the in vitro data presented are interesting and the novel structural data provided in this study do support the authors' contention that the unique flexibility of the N-domain drug binding site of fungal Hsp90 supports development of a species-specific agent, most likely by modifying an existing inhibitor.

Concerns:

1. I could not find the figure legends in any of the review documents provided, making clear interpretation of the figures somewhat difficult.
2. Fig. 5C: radicicol has a significant preference for human Hsp90alpha/beta in comparison to the fungal chaperone; at the same time, the CMLD compound preference for fungal Hsp90 vs human Hsp90alpha does not appear to match the 4.5-fold binding selectivity. While the concentration range is shown to be between 0 and 40 uM, it would be helpful to identify intermediate concentrations to allow comparison.
3. Fig. 8b: Although CMLD013075 appears much less toxic to human cells compared to radicicol and ganetespi, a 50% reduction in the fungal burden (left panel) requires approximately 1 uM compound. At 1 uM, CMLD013075 also reduces human cell survival (right panel) by 50%, perhaps a 2-fold improvement over radicicol and ganetespi. Do the authors consider this to be acceptable host toxicity (assuming in vivo data are similar)?

Reviewer #2 (Remarks to the Author):

Title: Structural basis for species-selective targeting of Hsp90 in a pathogenic fungus
The submitted manuscript describes the development of Hsp90 inhibitors derived from the natural products, radicicol and monocillin I, which target fungal Hsp90 with a slight preference over the human hsp90 ortholog. The inhibitors presented in the current work provide an alternative approach for the treatment of fungal infections, which are progressively exhibiting resistance to current antifungal therapies. With the help of co-crystal structures, the authors validate the conformational difference between *C. albicans* and human Hsp90 NBD. The observed differences have been exploited to develop semisynthetic derivatives that target the flexible NBD pocket of *C. albicans* Hsp90 selectively.

Observations and comments

- The lead compound CMLD013075 should be clearly identified in the figure 3
- To determine selectivity of the inhibitors, HEPG2 lysates have been used. Being a cancer cell line, it will contain Hsp90 in its active heteroprotein complex form. Hsp90 N-terminal inhibitors are known to exhibit preference for the cancer derived Hsp90 complex over the normal cell lysate derived Hsp90 complex. Therefore, the developed inhibitors are likely to display higher selectivity

if compared with the normal cell Hsp90 complex-this experiment should be completed.

- The observed *C. albicans* Hsp90 conformation in complex with CMLD013075 resembles the structure of Grp94 co-crystallized with PU-WS13 (PDB code: 3O2F) rather than the ADP complex. As a consequence, It will be important to determine the affinity of the lead compound against Grp94-this experiment should be performed.
- Color schemes selection in Fig 1 and 2 that differentiate between the human and fungal protein structures and ligands are not contrasting and difficult to distinguish.
- "Nucleotide binding" in place of binding in "chemo-structural evidence.. " section.
- CMLD013075 has been mentioned as CMLD010375 once.
- While KF58333 is an E form of oxime, the isomeric forms of the developed agents have not been mentioned. It seems that the crystallized form of CMLD013075 is the Syn form of oxime and evaluation of specific isomer might lead to improved fungal Hsp90 inhibition.

Reviewer #3 (Remarks to the Author):

This manuscript examines the differences between human and *Candida albicans* Hsp90s with the goal of developing inhibitors that target the fungal homolog. Using structural, chemical, and biological approaches, they report the discovery of a Radicicol-like compound that exhibits modest selectivity for the *Candida* homolog in binding and functional assays. Interestingly, they also find that the *Candida* Hsp90 N-terminal nucleotide binding domain exhibits greater plasticity in response to the binding of some ligands than the human variant.

The manuscript is difficult to evaluate due to the lack of legends for any of the figures, both main or supplemental, as well as the absence of any experimental methods. These should be added to any revised manuscript.

Within the limitations imposed by the lack of a complete manuscript, the following are concerns:

1. IC50s derived from FP competition binding measurements are not very accurate when comparing different proteins. In order to assure the accuracy of their binding data and to make a better correlation with their structural data, I would recommend that the authors repeat the FP binding studies using purified *Candida* and human Hsp90 proteins and compare the resulting K_i 's after determining the affinity of the tracer for each homolog. Alternatively, the authors could use ITC to measure the K_d 's for each ligand directly. Since the correlation between structural variations in the two homologs and the binding data is a central element of the manuscript, the binding data should be as accurate as possible.
2. The functional data reported by the authors in Figures 3d, 3e, 5c, 6a, and 8a shows extremely sharp cutoffs for inhibitor effects on organism growth. Perhaps the inhibitor concentration scale is not linear, although this is not mentioned anywhere in the manuscript.
3. There does not seem to be much correlation between the affinity of an inhibitor and its functional effect. It would help if the authors could discuss this point in greater detail, focusing on how to distinguish secondary effects like efflux from effects due to inhibition of Hsp90.
4. The authors note unusual Hsp90 conformations for the *Candida*:AUY and *Candida*:CLMD complexes. Perhaps these could be compared to some of the other reported structures for Grp94 and Trap-1, which appear to have greater structural variation, to see if there is any similarity.

October 19, 2018

We thank the reviewers for their careful consideration of our manuscript. We greatly appreciate their efforts and constructive suggestions. By addressing their concerns and recommendations with new data and new discussion as detailed below, we think the manuscript is much improved. All changes to the text in response to review have been highlighted in yellow. We hope that the work will now be considered suitable for publication.

Point by Point Response

Reviewer #1 (Remarks to the Author):

This report by Whitesell et al. provides proof-of-concept data that support the feasibility of developing a fungal-specific Hsp90 inhibitor with minimal ability to inhibit host (human) Hsp90. Highlighting the importance of this work, these and other authors have previously shown that pharmacologic inhibition of fungal Hsp90 using inhibitors developed against the human protein provides a unique means of countering fungal resistance to standard of care agents. As part of this study, the authors found that the N-domain drug binding region is more flexible compared to that of human Hsp90, suggesting more opportunities to modify existing inhibitors to take advantage of this conformational flexibility in order to design a species-specific inhibitor. The authors report identification of such an inhibitor, a derivative of the natural product Hsp90 inhibitor radicicol, with a 4.5-fold fungal binding selectivity. Interestingly, they show that the clinically evaluated Hsp90 inhibitor SNX-2112 also displays an almost 4-fold higher affinity for fungal Hsp90 than for human Hsp90, suggesting that modification of this chemotype may provide another route to develop a fungal-specific inhibitor with preferred in vivo properties. While the preliminary attempts at in vivo evaluation of CMLD013075 were disappointing due to its poor PK, etc., the in vitro data presented are interesting and the novel structural data provided in this study do support the authors' contention that the unique flexibility of the N-domain drug binding site of fungal Hsp90 supports development of a species-specific agent, most likely by modifying an existing inhibitor.

Concerns:

1. *I could not find the figure legends in any of the review documents provided, making clear interpretation of the figures somewhat difficult.*

Response: We sincerely apologize for the omission. It appears an incomplete "Zip of files for Reviewer" was provided. We have corrected the error in the re-submission. Complete figure legends and methods are now provided.

2. *Fig. 5C: radicicol has a significant preference for human Hsp90alpha/beta in comparison to the fungal chaperone; at the same time, the CMLD compound preference for fungal Hsp90 vs human Hsp90alpha does not appear to match the 4.5-fold binding selectivity. While the concentration range is shown to be between 0 and 40 uM, it would be helpful to identify intermediate concentrations to allow comparison.*

Response: Intermediate concentrations have now been indicated for the heat maps in Fig 5C.

3. *Fig. 8b: Although CMLD013075 appears much less toxic to human cells compared to radicicol and*

ganetespi, a 50% reduction in the fungal burden (left panel) requires approximately 1 μ M compound. At 1 μ M, CMLD013075 also reduces human cell survival (right panel) by 50%, perhaps a 2-fold improvement over radicicol and ganetespi. Do the authors consider this to be acceptable host toxicity (assuming *in vivo* data are similar)?

Response: We do not consider a two-fold improvement in host toxicity acceptable, but we do not expect toxicity to rapidly proliferating cancer cells in culture to translate directly to host toxicity *in vivo*. In response to Query 2 from Reviewer #2 regarding the effects of Hsp90 inhibitors in non-cancer cell lines, we now provide cytotoxicity data for 3T3 mouse fibroblasts (Figure S7a). The cytotoxicity IC₅₀ value for CMLD013075 against these immortalized, but non-transformed cells is higher, but further optimization to improve whole cell selectivity would still clearly be desirable.

Reviewer #2 (Remarks to the Author):

Title: Structural basis for species-selective targeting of Hsp90 in a pathogenic fungus

The submitted manuscript describes the development of Hsp90 inhibitors derived from the natural products, radicicol and monocillin I, which target fungal Hsp90 with a slight preference over the human hsp90 ortholog. The Inhibitors presented in the current work provide an alternative approach for the treatment of fungal infections, which are progressively exhibiting resistance to current antifungal therapies. With the help of co-crystal structures, the authors validate the conformational difference between *C. albicans* and human Hsp90 NBD. The observed differences have been exploited to develop semisynthetic derivatives that target the flexible NBD pocket of *C. albicans* Hsp90 selectively.

Observations and comments

1. *The lead compound CMLD013075 should be clearly identified in the figure 3*

Response: Figure 3 has been modified to indicate that structure **1** is the lead compound CMLD 013075

2. *To determine selectivity of the inhibitors, HEPG2 lysates have been used. Being a cancer cell line, it will contain Hsp90 in its active heteroprotein complex form. Hsp90 N-terminal inhibitors are known to exhibit preference for the cancer-derived Hsp90 complex over the normal cell lysate derived Hsp90 complex. Therefore, the developed inhibitors are likely to display higher selectivity if compared with the normal cell Hsp90 complex-this experiment should be completed.*

Response: Whole cell cytotoxicity data and fluorescence polarization data are now provided for immortalized, but non-malignant mouse embryonic fibroblasts (3T3 cells) in Figure S7a and 7b. Radicicol and CMLD013075 are both less toxic to these cells than cancer-derived HEPG2 cells, but their potency relative to one another remains the same at \sim 10-fold. As the reviewer suggests, however, the reduced toxicity to 3T3 cells of CMLD013075 does lead to higher fungal selectivity for the compound. Consistent with whole cell data, the apparent binding affinity of inhibitors is lower in 3T3 cell lysate than in HEPG2 lysate (Figure S7b). The interpretation of these results is problematic, however, because the affinity for geldanamycin (upon which our FP probe is based) also varies between the lysates. A discussion of these new results and how we interpret them is now provided in the manuscript (p 15, highlighted in yellow).

3. *The observed C. albicans Hsp90 conformation in complex with CMLD013075 resembles the structure of Grp94 co-crystallized with PU-WS13 (PDB code: 3O2F) rather than the ADP complex. As a*

consequence, it will be important to determine the affinity of the lead compound against Grp94-this experiment should be performed.

Response: The affinity of CMLD013075 for recombinant fungal and human Hsp90 NBD as well human Grp-94 NBD as measured by fluorescence polarization is now presented in Figure 4c. Corresponding data for geldanamycin, SNX-2112 and AUY922 are presented in Fig 3b. In addition to generating these data, we also overlaid the *C. albicans*-CMLD013075 and the Grp94-PUWS13 complexes as suggested by the reviewer. The two complexes showed very limited shared structural similarities (r.m.s.d. 8.39Å main-chain atoms). The comparison with Grp94-ADP (r.m.s.d. 4.337Å main-chain atoms) referenced in the text was included simply to highlight similar rearrangement of the lid region observed upon binding of the phosphonucleotide. A discussion of these issues is now provided in the manuscript on p 21-22 (highlighted in yellow).

4. *Color schemes selection in Fig 1 and 2 that differentiate between the human and fungal protein structures and ligands are not contrasting and difficult to distinguish.*

Response: The colors used to depict structures and ligands have been modified to increase contrast between conformations in the unbound and bound state in the human vs. fungal structures.

5. *“Nucleotide binding” in place of binding in “chemo-structural evidence..” section.*

Response: The text has been corrected.

6. *CMLD013075 has been mentioned as CMLD010375 once.*

Response: The text has been corrected.

7. *While KF58333 is an E form of oxime, the isomeric forms of the developed agents have not been mentioned. It seems that the crystallized form of CMLD013075 is the Syn form of oxime and evaluation of specific isomer might lead to improved fungal Hsp90 inhibition.*

Response: We thank the reviewer for this astute observation. The Supplementary Methods section describes all oximes as mixtures of E/Z isomers. KF5833 was obtained from Kyowa Hakko Kogyo (subsequently acquired by Kirin holdings to become Kyowa Hakko Kirin) and is reported in the literature to be exclusively the E isomer (Soga, S., Sharma, S.V., Shiotsu, Y. et al. *Cancer Chemother Pharmacol* (2001) 48: 435. <https://doi.org/10.1007/s002800100373>). In our hands, however, we have observed the E/Z ratio for isomer-enriched samples of our radicicol and monocillin oxime derivatives to equilibrate in DMSO solution over time. An example of one such equilibration experiment, and discussion of its implications in our assays, is included in the Supplementary Methods section.

To further clarify this point, we have expanded the Supplementary Methods discussion of this important topic to include the following:

“While we chromatographically obtained isomer-enriched samples of some oxime mixtures, the experiments employing these enriched samples are confounded by the use of DMSO as vehicle (due to the slow equilibration). All attempts to ascertain differences in Hsp90 inhibitory activity among the enriched samples have thus far been ambiguous, with roughly equipotent activities observed across the enriched samples. Because of these observations, combined with our inability resolve the exact oxime stereochemistry crystallographically at the obtainable resolution (one isomer was arbitrarily modeled into the structure, while the other also fits the experimental data well) we do not wish to over-speculate as to any isomeric preference for inhibition among the oxime mixtures”.

Reviewer #3 (Remarks to the Author):

This manuscript examines the differences between human and *Candida albicans* Hsp90s with the goal of developing inhibitors that target the fungal homolog. Using structural, chemical, and biological approaches, they report the discovery of a Radicicol-like compound that exhibits modest selectivity for the *Candida* homolog in binding and functional assays. Interestingly, they also find that the *Candida* Hsp90 N-terminal nucleotide binding domain exhibits greater plasticity in response to the binding of some ligands than the human variant.

The manuscript is difficult to evaluate due to the lack of legends for any of the figures, both main or supplemental, as well as the absence of any experimental methods. These should be added to any revised manuscript.

Response: We sincerely apologize for the omission. It appears an incomplete “Zip of files for Reviewer” was provided. We have corrected the error in the re-submission. Complete figure legends and methods are now provided.

Within the limitations imposed by the lack of a complete manuscript, the following are concerns:

1. *IC50s derived from FP competition binding measurements are not very accurate when comparing different proteins. In order to assure the accuracy of their binding data and to make a better correlation with their structural data, I would recommend that the authors repeat the FP binding studies using purified *Candida* and human Hsp90 proteins and compare the resulting K_i 's after determining the affinity of the tracer for each homolog. Alternatively, the authors could use ITC to measure the K_d 's for each ligand directly. Since the correlation between structural variations in the two homologs and the binding data is a central element of the manuscript, the binding data should be as accurate as possible.*

Response: In response to this reviewer's recommendation, FP data have now been obtained using purified nucleotide binding domains derived from human and *C. albicans* Hsp90 as well as Human Grp94. Using saturation binding and equilibrium competition methods applicable to FP as described by Rossi and Taylor (manuscript reference #32), we were able to calculate K_d values for our ligands as recommended. Results are presented in our revised manuscript in Figures 2a, 2b, 4c and Figure S3a, 3b.

2. *The functional data reported by the authors in Figures 3d, 3e, 5c, 6a, and 8a shows extremely sharp cut-offs for inhibitor effects on organism growth. Perhaps the inhibitor concentration scale is not linear, although this is not mentioned anywhere in the manuscript.*

Response: Inhibitor concentration scales in functional assays are 2-fold serial dilutions. Specific concentrations are now indicated in the figures and clarification is provided in the figure legends

3. *There does not seem to be much correlation between the affinity of an inhibitor and its functional effect. It would help if the authors could discuss this point in greater detail, focusing on how to distinguish secondary effects like efflux from effects due to inhibition of Hsp90.*

Response: A discussion of the importance of compound uptake and retention by *C. albicans* is now included in the text on p22-23 and relevant references provided.

4. *The authors note unusual Hsp90 conformations for the *Candida*:AUY and *Candida*:CLMD complexes. Perhaps these could be compared to some of the other reported structures for Grp94 and Trap-1, which appear to have greater structural variation, to see if there is any similarity.*

Response: Structural overlays between *C. albicans* Hsp90 NBD in complex with AUY922 or CMLD013075 and Grp94 or TRAP-1 showed very limited similarities. The overlay between the fungal complexes and Grp94 structures are described in detail in our response to reviewer 2, point #3. Structural superposition between the above-mentioned fungal complexes and the TRAP-1 structure (pdb code 4IVG) showed large structural differences with r.m.s.d. of 9.6Å and 8.6Å over main chain atoms with the *C. albicans*: AUY922 and *C. albicans*: CMLD013075 complexes respectively. Even if only the more similar part of the structures is included in the comparison (~160 residues from $\alpha 2$, $\alpha 8$, $\alpha 9$, $\beta 2$, $\beta 3$ and $\beta 4$), the structural differences are still considerable with r.m.s.d. values of 1.4Å and 1.5Å over C $_{\alpha}$ atoms. These large differences might be explained by the fact that the nucleotide binding domains in all available TRAP-1 structures are dimerized.

Reviewers' comments:

Reviewer #2 (Remarks to the Author):

The authors have attempted to address all of my concerns, and the manuscript is now suitable for publication...

Reviewer #3 (Remarks to the Author):

This manuscript describes the characterization of the *Candida albicans* Hsp90 Nucleotide binding domain with the goal of exploiting differences between it and the NBD of human Hsp90 for the development of *Candida* selective inhibitors. The investigators have identified conformational changes of the *Candida* Hsp90 in response to some classes of Hsp90 inhibitors and have tried to exploit these regions of conformational flexibility in the design of *Candida* selective inhibitors.

This is an improved version of the previous manuscript, which, among other things, lacked figure legends and a methods section. In addition to adding these important sections, the authors have done a good job responding to the earlier critiques.

Below are a few weaknesses in the technical descriptions of some of the experiments in the current manuscript that should be corrected before publication.

1. Ligand binding. A key finding of the manuscript is that the CLMD013075 ligand is selective for the *Candida* ortholog. The authors employ a fluorescence polarization competitive displacement assay to evaluate the binding properties of the CLMD013075 ligand to the various Hsp90s. Unfortunately, this is a less accurate and informative assay compared to other approaches. While FP-comp is consistent when applied to the same protein, the accuracy when used to compare the binding across Hsp90 orthologs is lower, especially when the fold selectivity is a log or less, as in the present case. While not rising to the level of a requirement for publication, I think it is worth pointing out that it is a missed opportunity that the K_d values and associated thermodynamic or kinetic parameters for CLMD binding to the various orthologs have not been definitively determined by ITC or SPR.

Specific points:

- a. The description of the FP experiment in the methods section is incomplete. Please include the concentration of the tracer ligand as well as the concentration of each purified protein.
- b. The description of the data analysis, and in particular the calculation of K_i from the IC_{50} , is still ambiguous. Ref. 32 cites both Cheng-Prusoff as well as Nikolovska-Coleska (Anal. Biochem. 2004) for this treatment. If the more accurate N-C method was used, and hopefully it was, please add the N-C reference to the references.
- c. In this regard, it would be helpful if the IC_{50} , which is the measured value in the FP-comp experiment, could be listed along with the derived K_i . This table could be added to the supplementary data.
- d. To avoid ambiguity, the authors should replace "KD" or "inhibitory KD" with " K_i ," when referring to the results of the FP competition assays, which is a more accurate term when describing a competition binding experiment.
- e. The manuscript states that each experiment was repeated but no error estimates are given for any of the K_i values. These should be added to each K_i value.

2. Line 592. Crystallographic experiment description. Please include the temperature at which the crystals were grown.

3. Structure descriptions.

- a. In some of the crystal structures of *Candida* Hsp90 strand 1 is peeled off of strand 6 (Figures

2b, 2c, 2d, 4a). It would be helpful if the authors could please describe whether or not this peeled off strand is stabilized by a packing interaction with another protein molecule in the crystal lattice. The crystallization conditions for each of the *Candida* Hsp90 NBD complexes are different and may have selected for a conformational variant that stabilizes lattice contacts.

b. Lines 46, 194, 196, 208, 224, 230, 458 and 489-491. The conformational changes the authors see in the *Candida* Hsp90 NBD in response to ligand binding are dramatic, stark, unlike anything seen before, remarkable, unprecedented, and extensive only if one does not reference the more extensive changes previously seen in the Trap-1 and Grp94 NBDs – other Hsp90 family members - in response to ligand binding, or the original yeast NBD crystal structure. It is also worthwhile keeping in mind that the relative lack of significant conformational rearrangements in human Hsp90 NBD crystal structures might reflect the ease with which the crystal packing arrangement that predominates in the PDB is achieved, rather than an inability of this NBD to make similar conformational changes in response to ligand binding. The authors make important and insightful structural comparisons in this report, but a better sense of perspective would be achieved if the authors would please consider removing the adjectival superlatives.

4. Lines 137-145. Terms such as “similar” and “somewhat lower” are too vague for this important comparison. Please support these terms by giving the values for the K_d s and catalytic efficiencies in the text rather than only referring to the supplemental figures.

Reviewer #2 (Remarks to the Author):

The authors have attempted to address all of my concerns, and the manuscript is now suitable for publication...

We are delighted to have satisfactorily addressed all concerns.

Reviewer #3

This is an improved version of the previous manuscript, which, among other things, lacked figure legends and a methods section. In addition to adding these important sections, the authors have done a good job responding to the earlier critiques.

We thank the reviewer for additional comments which have allowed us to further improve the rigor of our manuscript. Changes made in response to the specific critiques indicated below are indicated in the text by Cyan highlighting.

Below are a few weaknesses in the technical descriptions of some of the experiments in the current manuscript that should be corrected before publication.

1. Ligand binding. A key finding of the manuscript is that the CLMD013075 ligand is selective for the Candida ortholog. The authors employ a fluorescence polarization competitive displacement assay to evaluate the binding properties of the CLMD013075 ligand to the various Hsp90s. Unfortunately, this is a less accurate and informative assay compared to other approaches. While FP-comp is consistent when applied to the same protein, the accuracy when used to compare the binding across Hsp90 orthologs is lower, especially when the fold selectivity is a log or less, as in the present case. While not rising to the level of a requirement for publication, I think it is worth pointing out that it is a missed opportunity that the Kd values and associated thermodynamic or kinetic parameters for CLMD binding to the various orthologs have not been definitively determined by ITC or SPR.

We agree with this reviewer that providing affinity measurements determined by alternative biophysical techniques should not be a requirement for publication. Because performing ITC or SPR experiments would substantially delay publication of this work without materially changing its conclusions, we decline to do so in the current revision.

Specific points:

a. The description of the FP experiment in the methods section is incomplete. Please include the concentration of the tracer ligand as well as the concentration of each purified protein.

The concentration of tracer ligand (Cy3-geldanamycin) used for FP experiments was 0.1 nM and the concentration of recombinant purified proteins was 10 nM for human and yeast Hsp90 NBD and 7.5 nM for human Grp94. This information has been added to the methods section.

b. The description of the data analysis, and in particular the calculation of Ki from the IC50, is still

ambiguous. Ref. 32 cites both Cheng-Prusoff as well as Nikolovska-Coleska (Anal. Biochem. 2004) for this treatment. If the more accurate N-C method was used, and hopefully it was, please add the N-C reference to the references.

Calculation of K_i from FP EC50 data was performed as described in detail by Ref. 32 (Rossi and Taylor, Nature Protocols, 2011), which is why this reference was cited rather than the one indicated by this reviewer. Specifically, the calculation of K_d for our Cy3-GdA probe molecule in relation to each of the recombinant NBD (*C. albicans* Hsp90, Human Hsp90-beta and Human Grp94) was performed by solving the equations as laid out in **Box 1** provided in the reference cited. With K_d 's established for our probe, K_i values were then calculated from equilibrium competition data by solving the equations provided in **Box 2** of the same reference. As described in this reference, the specified method of analysis involves both the Cheng-Prusoff equation and the approach described by Nikolovska, Coleska et al. in the reference cited by this reviewer. We think the method of analysis we used provides the most rigorous and accurate approach to analysis of FP data currently available. The raw FP data and calculations performed on them are now provided in the Excel "Source Data File" accompanying this revised re-submission.

c. In this regard, it would be helpful if the IC50, which is the measured value in the FP-comp experiment, could be listed along with the derived K_i . This table could be added to the supplementary data.

Raw FP data, EC50 derived from them by curve fitting in Prism and calculations performed to generate K_i values are now provided in the Excel "Source Data File" accompanying this revised re-submission.

d. To avoid ambiguity, the authors should replace "KD" or "inhibitory KD" with " K_i ," when referring to the results of the FP competition assays, which is a more accurate term when describing a competition binding experiment.

" K_d " has been replaced by " K_i " when describing competition binding results.

e. The manuscript states that each experiment was repeated but no error estimates are given for any of the K_i values. These should be added to each K_i value.

Mean and standard error of the mean for the results from 3 independent experiments consisting of duplicate technical determinations are now provided for all K_i values reported.

2. Line 592. Crystallographic experiment description. Please include the temperature at which the crystals were grown.

The crystallization temperature of 21°C degrees is now included in methods

3. Structure descriptions.

a. In some of the crystal structures of Candida Hsp90 strand 1 is peeled off of strand 6 (Figures 2b, 2c, 2d, 4a). It would be helpful if the authors could please describe whether or not this peeled off strand is stabilized by a packing interaction with another protein molecule in the crystal lattice. The crystallization conditions for each of the Candida Hsp90 NBD complexes are different and may have selected for a conformational variant that stabilizes lattice contacts.

In the *C. albicans* Hsp90 NBD in complex with AUY922 structure, the N-terminal strand peels off and loses its extended conformation. As suggested by the reviewer and demonstrated in the figure below

(left panel), the crystal packing shows the N-terminal strand residues (7-ETHE-10) interacting with a symmetry molecule (-x,-y-1,z). While it is possible as suggested by the reviewer that the observed conformation of the fungal Hsp90-AUY922 complex is the result of the crystal packing, we see no clear evidence to suggest that the crystal contacts are the driving force behind the peeling off of the N-terminal strand. The crystal packing includes contacts with four symmetry molecules (-x,-y-1,z; -x+1/2,y,-z-1/4; -y,-x,-z and -x-1/2,y,-z-1/4). It buries 3,283Å² (27.6% of the protein surface) and the 10 N-terminal residues account for 10.9% (357Å²) of the crystal lattice interface area. Furthermore, crystal lattice contacts are absent in this region in the *C. albicans* Hsp90 NBD-CMLD013075 complex crystal (figure below, right panel) and this structure also shows the peeling off of the N-terminal strand (manuscript Figures 4a and S7b). Thus, we conclude that ligand binding is the most probable selective force behind the observed conformations of *C. albicans* Hsp90 NBD.

Contribution of crystal packing to observed structures: *Left panel:* VDW representation of AUY922 (cyan) in complex with *C. albicans* Hsp90 NBD (green). Residues from strand 1 are highlighted in red, and the preceding N-terminal residues are highlighted in orange. *Right panel:* VDW representation of CMLD013075 (magenta) in complex with *C. albicans* Hsp90 NBD (molecule A: green, molecule B cyan). N-terminal residues are highlighted in red (molecule A) and orange (molecule B).

b. Lines 46, 194, 196, 208, 224, 230, 458 and 489-491. The conformational changes the authors see in the Candida Hsp90 NBD in response to ligand binding are dramatic, stark, unlike anything seen before, remarkable, unprecedented, and extensive only if one does not reference the more extensive changes previously seen in the Trap-1 and Grp94 NBDs – other Hsp90 family members - in response to ligand binding, or the original yeast NBD crystal structure. It is also worthwhile keeping in mind that the relative lack of significant conformational rearrangements in human Hsp90 NBD crystal structures might reflect the ease with which the crystal packing arrangement that predominates in the PDB is achieved, rather than an inability of this NBD to make similar conformational changes in response to ligand binding. The authors make important and insightful structural comparisons in this report, but a better sense of perspective would be achieved if the authors would please consider removing the adjectival superlatives.

Adjectival superlatives have been deleted.

4. Lines 137-145. Terms such as “similar” and “somewhat lower” are too vague for this important comparison. Please support these terms by giving the values for the Kds and catalytic efficiencies in the text rather than only referring to the supplemental figures.

Specific values have been inserted into the text.